# Quantum Algorithms for Non-smooth Non-convex Optimization

**Chengchang Liu**[* 1]   **Chaowen Guan**[* 2]   **Jianhao He**[# 1]   **John C.S. Lui**[1]

[1] The Chinese University of Hong Kong   [2] University of Cincinnati

7liuchengchang@gmail.com   guance@ucmail.uc.edu
jianhaohe9@cuhk.edu.hk   cslui@cse.cuhk.edu.hk

## Abstract

This paper considers the problem of finding the $(\delta, \epsilon)$-Goldstein stationary point of the Lipschitz continuous objective, which is a rich function class to cover a large number of important applications. We construct a novel zeroth-order quantum estimator for the gradient of the smoothed surrogate. Based on such estimator, we propose a novel quantum algorithm that achieves a query complexity of $\tilde{\mathcal{O}}(d^{3/2}\delta^{-1}\epsilon^{-3})$ on the stochastic function value oracle, where $d$ is the dimension of the problem. We also improve the query complexity to $\tilde{\mathcal{O}}(d^{3/2}\delta^{-1}\epsilon^{-7/3})$ by introducing a variance reduction variant. Our findings demonstrate the clear advantages of using quantum techniques for non-convex non-smooth optimization, as they outperform the optimal classical methods in dependence on $\epsilon$ by a factor of $\epsilon^{-2/3}$.

## 1 Introduction

In this paper, we study the following problem

$$\min_{\mathbf{x} \in \mathbb{R}^d} \left\{ f(\mathbf{x}) \triangleq \mathbb{E}_\xi \left[ F(\mathbf{x}; \xi) \right] \right\}, \tag{1}$$

where the stochastic component $F(\mathbf{x}; \xi)$ is $L$-Lipschitz continuous but possibly *non-convex* and *non-smooth*. This problem has received increasing attention recently because it is general enough to cover many important applications, including deep neural networks [21, 40], reinforcement learning [9, 49], and statistical learning [17, 39, 62].

Due to the absence of both smoothness and convexity in the objective function, neither the gradient nor the sub-differentials are valid anymore to measure the convergence behavior. The Clarke subdifferential is a natural extension for describing the first-order information of the Lipschitz continuous function [10], however, it is intractable for finding the near-approximate stationary point in terms of the Clarke subdifferential as suggested by the hard instances [31, 50, 63]. Zhang et al. [63] introduce the notion of $(\delta, \epsilon)$-Goldstein stationary point (cf. Section 2.2), which weakens the traditional stationary point by considering the convex hull of the Clarke subdifferentials. Following this, we focus on the problem of finding the $(\delta, \epsilon)$-Goldstein stationary points of the objective.

There are many optimization methods for finding the $(\delta, \epsilon)$-Goldstein stationary points via classical stochasic oracles [6, 14, 28, 31, 35, 47, 52, 63]. Zhang et al. [63] proposed stochastic interpolated normalized gradient descent method (SINGD) with the first non-asymptotic result, which has the stochastic first-order complexity of $\mathcal{O}(\delta^{-1}\epsilon^{-4})$. Later, Tian et al. [52] developed the perturbed

---

[*]denotes equal contributions; [#] denotes the corresponding author.

38th Conference on Neural Information Processing Systems (NeurIPS 2024).

Table 1: We summarize the complexities of classical and quantum zeroth-order methods for finding the $(\epsilon, \delta)$-Goldstein point of a *non-smooth non-convex* objective, where $d$ is the dimension of the problem.

| Methods | Oracle | Query Complexity | Reference |
|---------|--------|------------------|-----------|
| GFM | classical | $\mathcal{O}\left(d^{3/2}\delta^{-1}\epsilon^{-4}\right)$ | Lin et al. [35] |
| GFM+ | classical | $\mathcal{O}\left(d^{3/2}\delta^{-1}\epsilon^{-3}\right)$ | Chen et al. [6] |
| OptimalZO | classical | $\mathcal{O}\left(d\delta^{-1}\epsilon^{-3}\right)$ | Kornowski and Shamir [32] |
| QGFM | quantum | $\tilde{\mathcal{O}}\left(d^{3/2}\delta^{-1}\epsilon^{-3}\right)$ | Theorem 4.1 |
| QGFM+ | quantum | $\tilde{\mathcal{O}}\left(d^{3/2}\delta^{-1}\epsilon^{-7/3}\right)$ | Theorem 4.3 |

Table 2: We summarize the complexities of classical and quantum first-order methods for finding the $\epsilon$-stationary point of a *smooth non-convex* objective, where $d$ is the dimension of the problem.

| Methods | Oracle | Query Complexity | Reference |
|---------|--------|------------------|-----------|
| SPIDER/PAGE | classical | $\mathcal{O}(\epsilon^{-3})$ | Fang et al. [18], Li et al. [34] |
| Q-SPIDER | quantum | $\tilde{\mathcal{O}}(d^{1/2}\epsilon^{-5/2})$ | Sidford and Zhang [48] |
| QGM+ | quantum | $\tilde{\mathcal{O}}(d^{1/2}\epsilon^{-7/3})$ | Theorem G.1 |

SINGD method which queries the gradient at the differentiable point and established the same complexity. Cutkosky et al. [13] improved the stochastic first-order oracle complexities to $\mathcal{O}(\delta^{-1}\epsilon^{-3})$ by using the "online to non-convex conversation", assuming $f(\cdot)$ is differentiable. This improvement aligns with the theoretical lower bound [13].

Zeroth-order methods, which only query the function value oracle, are more practical for the Lipschitz continuous objective. This is because computing first-order oracles can be extremely challenging [29, 52] or even inaccessible for numerous real-world applications [16, 27, 43]. Lin et al. [35] proposed a gradient-free method to find the $(\delta, \epsilon)$-Goldstein stationary point within $\mathcal{O}(d^{3/2}\delta^{-1}\epsilon^{-4})$ query complexity to the stochastic function value via a connection between the randomized smoothing [41] and the Goldstein stationary point. This complexity was further improved to $\mathcal{O}(d^{3/2}\delta^{-1}\epsilon^{-3})$ and $\mathcal{O}(d\delta^{-1}\epsilon^{-3})$ by Chen et al. [6], Kornowski and Shamir [32] respectively. However, all these methods using the classical oracles to find the Goldstein stationary point face a bottleneck of $\delta^{-1}\epsilon^{-3}$ due to the lower bound reported by [13].

Recently, we have witnessed the power of quantum optimization methods by accessing the quantum counterparts of classical oracles for *non-convex* optimization [7, 23, 37, 48, 61, 64], *convex* optimization [4, 5, 48, 55, 64], and semi-definite programming [1, 2, 53, 54]. However, most of these results focus on deterministic methods and the case where the objective function is *smooth*. Garg et al. [19] and Zhang and Li [60] showed the negative results for *non-smooth convex* and *smooth non-convex* optimization that quantum algorithms have no improved rates over classical ones when the dimension is large. Sidford and Zhang [48] proposed stochastic quantum methods which show the advantage of using quantum stochastic first-order oracles for *smooth* objectives when the dimension is relatively small. To the best of our knowledge, there is no work showing the quantum speedups for minimizing *non-smooth non-convex* objectives, which is the most general and fundamental function class. Based on this, it is a natural question to ask:

*Can we go beyond the complexity of $\mathcal{O}(\delta^{-1}\epsilon^{-3})$ to find the $(\delta, \epsilon)$-Goldstein stationary point for non-smooth non-convex stochastic optimization by involving quantum oracles?*

We give an affirmative answer to the above question by proposing novel quantum zeroth-order methods and showing their explicit query complexities. We summarize our contributions as follows.

- We construct efficient quantum gradient estimators for the smoothed surrogate of the objectives with $\mathcal{O}(1)$-queries of the function value oracles, which allows us to construct efficient quantum

zeroth-order methods. Moreover, we provide explicit constructions of quantum superposition over required distributions. We present these results in Section 3 and Appendix A.

- We propose the quantum gradient-free method (QGFM) and the fast quantum gradient-free method (QGFM+) for *non-smooth non-convex* optimization. We achieve the query complexities of $\mathcal{O}(d^{3/2}\delta^{-1}\epsilon^{-3})$ for QGFM and $\mathcal{O}(d^{3/2}\delta^{-1}\epsilon^{-7/3})$ for QGFM+ in finding the $(\delta, \epsilon)$-Goldstein stationary point using quantum stochastic function value oracle. The query complexity of QGFM+ surpasses the optimal result achieved by classical methods by a factor of $\epsilon^{-2/3}$. We compare our methods with the classical zeroth-order methods in Table 1 and present the results in Section 4.

- We generalize the algorithm framework of QGFM+ for *smooth non-convex* optimization (i.e. the gradient of the objective function is Lipschitz continuous). We propose the fast quantum gradient method (QGM+), which takes the advantage of QGFM+ to choose the variance level adaptively. QGM+ enjoys an improved complexity of $\tilde{\mathcal{O}}(d^{1/2}\epsilon^{-7/3})$ queries of the quantum stochastic gradient oracle, which outperforms the existing state-of-the-art method (Q-SPIDER [48]) by a factor of $\epsilon^{-1/6}$. We compare our method with the classical and quantum first-order methods in Table 2. A discussion of this is presented in Remark 4.5, and the formal results are stated in Appendix G.

## 2 Preliminaries

We introduce preliminaries for quantum computing model and non-smooth non-convex optimization in this section.

### 2.1 Preliminaries for Quantum Computing Model

Here we formally review the basics and some concepts from quantum computing with which we work. For more details, see Nielsen and Chuang [42].

**Quantum Basics.** A quantum state can be seen as a vector $\mathbf{x} = (x_1, x_2, \ldots, x_m)^\top$ in the Hilbert space $\mathcal{H}^m$ such that $\sum_i |x_i|^2 = 1$. We follow the Dirac bra/ket notation on quantum states, i.e., we denote the quantum state for $\mathbf{x}$ by $|\mathbf{x}\rangle$ and denote $\mathbf{x}^\dagger$ by $\langle\mathbf{x}|$, where $\dagger$ means the Hermitian conjugation.

Given a state $|\psi\rangle = \sum_{i=1}^m c_i|i\rangle$, we call $c_i \in \mathbb{C}$ the amplitude of the state $|i\rangle$. Given two quantum states $|\mathbf{x}\rangle \in \mathcal{H}^m$ and $|\mathbf{y}\rangle \in \mathcal{H}^m$, we denote their inner product by $\langle\mathbf{x}|\mathbf{y}\rangle \triangleq \sum_i x_i^\dagger y_i$. Given $|\mathbf{x}\rangle \in \mathcal{H}^m$ and $|\mathbf{y}\rangle \in \mathcal{H}^n$, we denote their tensor product by $|\mathbf{x}\rangle \otimes |\mathbf{y}\rangle \triangleq (x_1y_1, \cdots, x_my_n)^\top \in \mathcal{H}^{m\times n}$. If we measure state $|\psi\rangle = \sum_{i=1}^m c_i|i\rangle$ on a computational basis, we will obtain $i$ with probability $|c_i|^2$ and the state will collapse into $|i\rangle$ after measurement for all $i$. A quantum algorithm works by applying a sequence of unitary operators to a initial quantum state.

**Quantum Query Complexity.** Corresponding to the classical query model, quantum query complexity considers the number of queries to a black box of a particular function which needs to be invoked to solve a problem. In many cases, the black box corresponds to the process that has the highest overhead, and therefore reducing the number of queries to it will effectively reduce the computational complexity of the entire algorithm. For example, if a classical oracle $\mathbf{C}_f$ for a function $f$ is a black box that, when queried with a point $\mathbf{x}$, outputs the function value $\mathbf{C}_{f(\mathbf{x})} = f(\mathbf{x})$, then the corresponding quantum oracle $\mathbf{U}_f$ is a unitary transformation that maps a quantum state $|\mathbf{x}\rangle|q\rangle$ to the state $|\mathbf{x}\rangle|q + f(\mathbf{x})\rangle$. Moreover, given the superposition input $\sum_{\mathbf{x},q} \alpha_{\mathbf{x},q}|\mathbf{x}\rangle|q\rangle$, applying the quantum oracle once will, by linearity, output the quantum state $\sum_{\mathbf{x},q} \alpha_{\mathbf{x},q}|\mathbf{x}\rangle|q + f(\mathbf{x})\rangle$.

### 2.2 Preliminaries for Non-convex Non-smooth Optimization

We introduce the necessary background for non-convex non-smooth optimization, with the following mild assumption that the objective function is Lipschitz continuous.

**Assumption 1.** *We assume the stochastic component $F(\cdot; \xi)$ of the objective $f(\cdot)$ satisfies that $|F(\mathbf{x}; \xi) - F(\mathbf{y}; \xi)| \leq L\|\mathbf{x} - \mathbf{y}\|$ for every $\mathbf{x}, \mathbf{y} \in \mathbb{R}^d$. In addition, we assume $f : \mathbb{R}^d \to \mathbb{R}$ is lower bounded and denote $f^* \triangleq \inf_{\mathbf{x}\in\mathbb{R}^d} f(\mathbf{x})$.*

The Rademencher theorem indicates that $f(\cdot)$ is differentiable almost everywhere under Assumption 1, which allows us to define its Clarke subdifferential as follows [10].

**Definition 2.1** (Clarke sub-differential). *The Clarke sub-differential of a Lipschitz function at point* $\mathbf{x}$ *is defined by* $\partial f(\mathbf{x}) \triangleq \text{conv} \{\mathbf{g} : \mathbf{g} = \lim_{\mathbf{x}_k \to \mathbf{x}} \nabla f(\mathbf{x}_k)\}$.

We then introduce the Goldstein subdifferential [22] and the $(\delta, \epsilon)$-Goldstein stationary point [63].

**Definition 2.2** (Goldstein sub-differential). *The Goldstein subdifferential of a Lipschitz function at point* $\mathbf{x}$ *is defined by* $\partial_\delta f(\mathbf{x}) \triangleq \text{conv} \{\cup_{\mathbf{y} \in \mathbf{B}_\delta(\mathbf{x})} \partial f(\mathbf{y})\}$.

**Definition 2.3** ($(\delta, \epsilon)$-Goldstein stationary point). *We call* $\mathbf{x}$ *the* $(\delta, \epsilon)$-*Goldstein stationary point of a given Lipschitz function if it satisfies* $\text{dist}(0, \partial_\delta f(\mathbf{x})) \le \epsilon$, *where* $\partial_\delta f(\mathbf{x})$ *is the Goldstein subdifferential.*

Next, we define the smoothed surrogate of $f(\cdot)$ as follows.

**Definition 2.4** ($\delta$-smoothed surrogate). *The* $\delta$-*smoothed surrogate of* $f$ *is defined by*

$$f_\delta(\mathbf{x}) \triangleq \mathbb{E}_{\mathbf{w} \sim \mathcal{P}} \left[ f(\mathbf{x} + \delta \mathbf{w}) \right], \tag{2}$$

*where* $\mathcal{P}$ *is the uniform distribution on a unit ball.*

Although $f(\cdot)$ is non-smooth, its smoothed surrogate $f_\delta(\cdot)$ enjoys some good properties as presented in the following proposition [6, 15, 35, 59].

**Proposition 2.1.** *If* $f(\cdot)$ *satisfies Assumption 1, its smoothed surrogate* $f_\delta(\cdot)$ *satisfies that:*

- $|f_\delta(\cdot) - f(\cdot)| \le \delta L$ *and* $|f_\delta(\mathbf{x}) - f_\delta(\mathbf{y})| \le L \|\mathbf{x} - \mathbf{y}\|$.

- $\nabla f_\delta(\cdot)$ *is* $c\sqrt{d}L\delta^{-1}$-*Lipschitz for some constant* $c > 0$, *i.e.* $\|\nabla f_\delta(\mathbf{x}) - \nabla f_\delta(\mathbf{y})\| \le c\sqrt{d}L\|\mathbf{x} - \mathbf{y}\|$.

- $\nabla f_\delta(\cdot) \in \partial_\delta f(\cdot)$, *where* $\partial_\delta f(\cdot)$ *is the Goldstein subdifferential.*

*Remark* 2.2. Proposition 2.1 implies that the task of finding the $(\delta, \epsilon)$-Goldstein stationary point of $f(\cdot)$ is equivalent to finding the $\epsilon$-stationary point of a smoothed function $f_\delta(\cdot)$, i.e. finding some point $\mathbf{x}$ such that $\|\nabla f_\delta(\mathbf{x})\| \le \epsilon$.

## 3 Zeroth-order Based Stochastic Quantum Estimator

In this section, we present a novel quantum estimator for the gradient of the smoothed surrogate $f_\delta(\cdot)$ by using the quantum stochastic function value oracle, which is essential for designing our quantum algorithms for non-convex non-smooth optimization.

### 3.1 Quantum Estimators via Quantum Stochastic Function Value Oracle

In this section, we construct quantum estimators for the gradient of the smoothed surrogate by $\mathcal{O}(1)$ -queries of the quantum stochastic function value oracle.

We start with the definition of the stochastic function value oracle. Classically, a stochastic function value evaluation is defined as $F(\mathbf{x}, \xi)$ for a function $f : \mathbb{R}^d \to \mathbb{R}$ with $\xi$ such that $\mathbb{E}_\xi[F(\mathbf{x}, \xi)] = f(\mathbf{x})$. In this work, we assume access to a *quantum* stochastic function value oracle $\mathbf{U}_F$ for $f(\cdot)$, which is defined as follows.

**Definition 3.1** (Quantum stochastic function value oracle). *For* $f : \mathbb{R}^d \to \mathbb{R}$, *the quantum stochastic function value oracle, denoted by* $\mathbf{U}_F$, *works as:* $\mathbf{U}_F : |\mathbf{x}\rangle \otimes |\xi\rangle \otimes |b\rangle \longmapsto |\mathbf{x}\rangle \otimes |\xi\rangle \otimes |b + F(\mathbf{x}, \xi)\rangle$, *where* $F(\mathbf{x}, \xi)$ *is sampled from a distribution* $p_\xi(\cdot)$ *such that* $\mathbb{E}_\xi[F(\mathbf{x}; \xi)] = F(\mathbf{x})$.

It is common to construct the following stochastic gradient estimator for $\nabla f_\delta(\cdot)$ [6, 32, 35, 36, 41]:

$$\mathbf{g}_\delta(\mathbf{x}; \mathbf{w}, \xi) \triangleq \frac{d}{2\delta} \left( F(\mathbf{x} + \delta \mathbf{w}; \xi) - F(\mathbf{x} - \delta \mathbf{w}; \xi) \right) \cdot \mathbf{w}, \tag{3}$$

where $\mathbf{w} \in \mathbb{R}^d$ is uniformly distributed on a unit sphere. The following proposition shows that $\mathbf{g}_\delta(\mathbf{x}; \mathbf{w}, \xi)$ is a good estimator of $\nabla f_\delta(\cdot)$.

**Proposition 3.1** ([6, Proposition 3 and 4]). *Under Assumption 1, i.e. the random variable $\xi$ satisfies that*

$$|F(\mathbf{x};\xi) - F(\mathbf{y};\xi)| \le L\|\mathbf{x} - \mathbf{y}\| \quad and \quad \mathbb{E}_\xi[F(\mathbf{x};\xi)] = f(\mathbf{x}), \tag{4}$$

*hold for all $\mathbf{x}, \mathbf{y} \in \mathbb{R}^d$, then $\mathbf{g}_\delta(\mathbf{x};\mathbf{w},\xi)$ defined in eq. (3) satisfies that $\mathbb{E}_{\mathbf{w},\xi}[\mathbf{g}_\delta(\mathbf{x};\mathbf{w},\xi)] = \nabla f_\delta(\mathbf{x})$, $\mathbb{E}_{\mathbf{w},\xi}[\|\mathbf{g}_\delta(\mathbf{x};\mathbf{w},\xi) - \nabla f_\delta(\mathbf{x})\|^2] \le c\pi dL^2$, and $\mathbb{E}_{\mathbf{w},\xi}[\|\mathbf{g}_\delta(\mathbf{x};\mathbf{w},\xi) - \mathbf{g}_\delta(\mathbf{y};\mathbf{w},\xi)\|^2 \le \frac{d^2 L^2}{\delta^2}\|\mathbf{x} - \mathbf{y}\|^2$, where $c = 16\sqrt{2}\pi$.*

Next, to exploit the power of quantum algorithms, we generalize eq. (3) to its quantum counterpart. Based on eq. (3) and Proposition 3.1, $\mathbf{g}_\delta(\mathbf{x};\mathbf{w},\xi)$ can be interpreted as a random variable. In the quantum setting, accessing a random variable typically involves querying a *quantum sampling oracle*, which returns a quantum superposition over the associated distribution.

**Definition 3.2** (Quantum sampling oracle). *For a random variable $X$ with sample space $\Omega$, its quantum sampling oracle $\mathbf{O}_X$ is defined as $\mathbf{O}_X : |0\rangle \longmapsto \sum_{\mathbf{x}} \sqrt{\Pr[X = \mathbf{x}]}|\mathbf{x}\rangle \otimes |\psi_{\mathbf{x}}\rangle$, where $|\psi_{\mathbf{x}}\rangle$ is an arbitrary quantum state for every $\mathbf{x}$.*

The content in the second quantum register can also be viewed as possible quantum garbage appearing during the implementation of the oracle. Observe that if we directly measure the output of $\mathbf{O}_X$, it will collapse to a classical sampling access to $X$ that returns a random sample $\mathbf{x}$ with respect to probability $\Pr[X = \mathbf{x}]$. Note that the output of $\mathbf{O}_X$ can also be represented as integral to continuous random variables, as used in [8, 48].

Hence, based on our observation that $\mathbf{g}_\delta(\mathbf{x};\mathbf{w},\xi)$ can be viewed as a random variable, our target oracle $\mathbf{O}_{\mathbf{g}_\delta}$–quantum stochastic gradient oracle–is essentially a quantum sampling oracle. Given this, we formally define the quantum $\delta$-estimated stochastic gradient oracle as follows.

**Definition 3.3** (Quantum $\delta$-estimated stochastic gradient oracle). *For $f_\delta(\cdot) : \mathbb{R}^d \to \mathbb{R}$, its quantum $\delta$-estimated stochastic gradient oracle is defined as*

$$\mathbf{O}_{\mathbf{g}_\delta} : |\mathbf{x}\rangle \otimes |\mathbf{0}\rangle \otimes |\mathbf{0}\rangle \longmapsto |\mathbf{x}\rangle \otimes \sum_{\xi,\mathbf{w}} \sqrt{\Pr[\mathbf{w},\xi]}|\mathbf{g}_\delta(\mathbf{x};\mathbf{w},\xi)\rangle \otimes |\psi_{\mathbf{w},\xi}\rangle,$$

*where the random variable $\mathbf{w}$ is uniformly distributed in a unit sphere and $\xi$ satisfies eq. (4).*

Proposition 3.1 implies $\mathbf{g}_\delta(\cdot)$ can serve as an estimator of $\nabla f_\delta$, and can be calculated with access to a quantum $\delta$-estimated stochastic gradient oracle as defined above. The following theorem shows that such an oracle can be built with only $\mathcal{O}(1)$ access to the quantum stochastic function value oracle.

**Lemma 3.2.** *Given access to a quantum sampling oracle $\mathbf{O}_{\xi,\mathbf{w}}$ to the joint distribution on $(\xi, \mathbf{w})$, one can construct a quantum $\delta$-estimated stochastic gradient oracle (as defined in Definition 3.3) with two queries to the quantum stochastic function value oracle $\mathbf{U}_F$.*

*Remark* 3.3. In Lemma 3.2, we assume a black-box access to quantum sampling oracle $\mathbf{O}_{\xi,\mathbf{w}}$ following Sidford and Zhang [48]. We present the explicit construction of this oracle in Appendix A.

Similarly, we can also constructed the estimator of $\nabla f_\delta(\mathbf{x}) - \nabla f_\delta(\mathbf{y})$ by the following oracle:

$$\mathbf{O}_{\Delta\mathbf{g}_\delta} : |\mathbf{x}\rangle \otimes |\mathbf{y}\rangle \otimes |\mathbf{0}\rangle \otimes |\mathbf{0}\rangle \longmapsto |\mathbf{x}\rangle \otimes |\mathbf{y}\rangle \otimes \sum_{\xi,\mathbf{w}} \sqrt{\Pr[\mathbf{w},\xi]}|\mathbf{g}_\delta(\mathbf{x};\mathbf{w},\xi) - \mathbf{g}_\delta(\mathbf{y};\mathbf{w},\xi)\rangle \otimes |\psi_{\mathbf{w},\xi}\rangle,$$

with only $\mathcal{O}(1)$-queries of stochastic quantum function value oracle.

**Corollary 3.4.** *Under the same conditions as in Lemma 3.2, one can construct $\mathbf{O}_{\Delta\mathbf{g}_\delta}$ with four queries to the quantum stochastic function value oracle $\mathbf{U}_F$.*

### 3.2 Mini-batch Quantum Estimators via Quantum Mean Estimation

We constructed the quantum oracles $\mathbf{O}_{\mathbf{g}_\delta}$ and $\mathbf{O}_{\Delta\mathbf{g}_\delta}$ with $\mathcal{O}(1)$-queries of quantum function value oracles in Section 3.1. These oracles produce outputs in the form of random variables. Specifically, $\mathbf{O}_{\mathbf{g}_\delta}$ provides an output with expectation $\nabla f_\delta(\mathbf{x})$ with the input $\mathbf{x}$, and $\mathbf{O}_{\Delta\mathbf{g}_\delta}$ provides an output with expectation $\nabla f_\delta(\mathbf{x}) - \nabla f_\delta(\mathbf{y})$ for $\mathbf{O}_{\Delta\mathbf{g}_\delta}$ with the inputs $\mathbf{x}$ and $\mathbf{y}$.

---

**Algorithm 1** Quantum Gradient-Free Method (QGFM)

1: **for** $t = 0, 1 \ldots T$
2:      Construct $\mathbf{g}_t$ as an unbiased quantum estimator of $\nabla f_\delta(\mathbf{x}_t)$ with variance at most $\hat{\sigma}_t^2$ using $\mathbf{U}_F$ according to Theorem 3.5.
3:      $\mathbf{x}_{t+1} = \mathbf{x}_t - \eta \mathbf{g}_t$
4: **end for**

---

The variance of the outputs can be reduced by constructing the mini-batch estimator. Inspired by the recent advance on quantum mean estimation [11, 12, 48] which improve the classical mini-batch estimator for multi-dimensional random variables, we construct improved estimators for $\nabla f_\delta(\mathbf{x})$ and $\nabla f_\delta(\mathbf{x}) - \nabla f_\delta(\mathbf{y})$. We formally present the results in the following theorem.

**Theorem 3.5.** *Under Assumption 1, and given access to a quantum sampling oracle $\mathbf{O}_{\xi, \mathbf{w}}$ to the joint distribution on $(\xi, \mathbf{w})$, it holds that:*

1. *there exists an algorithm that can construct an unbiased quantum estimator $\hat{\mathbf{g}}$ of $\nabla f_\delta(\mathbf{x})$ such that $\mathbb{E}\left[\|\hat{\mathbf{g}} - \nabla f_\delta(\mathbf{x})\|^2\right] \le \hat{\sigma}_1^2$ within $\tilde{\mathcal{O}}(dL\hat{\sigma}_1^{-1})$ queries of $\mathbf{U}_F$ in expectation.*

2. *there exists an algorithm that can construct an unbiased quantum estimator $\Delta\mathbf{g}$ of $\nabla f_\delta(\mathbf{x}) - \nabla f_\delta(\mathbf{y})$ such that $\mathbb{E}\left[\|\Delta\mathbf{g} - (\nabla f_\delta(\mathbf{x}) - \nabla f_\delta(\mathbf{y}))\|^2\right] \le \hat{\sigma}_2^2$ within $\tilde{\mathcal{O}}(d^{3/2}L\|\mathbf{y} - \mathbf{x}\|\hat{\sigma}_2^{-1}\delta^{-1})$ queries of $\mathbf{U}_F$ in expectation.*

*Remark* 3.6. Compared to the classical mini-batch estimator for $\nabla f_\delta(\mathbf{x})$, which requires $\mathcal{O}(dL^2\hat{\sigma}_1^{-2})$ queries of $\mathbf{C}_F$ to achieve the level of variance $\hat{\sigma}_1^2$ ([6, Corollary 2.1]), our mini-batch quantum estimator for $\nabla f_\delta(\mathbf{x})$ in Theorem 3.5 reduces a factor of $L\hat{\sigma}_1^{-1}$ without increasing the dimension dependence.

## 4   Quantum Algorithms for Finding the Goldstein Stationary Point

In this section, we develop novel quantum algorithms for finding the $(\delta, \epsilon)$-Goldstein stationary point of a non-smooth non-convex objective $f(\cdot)$. Instead of finding the stationary point directly, we consider finding the $\epsilon$-stationary point of its smoothed surrogate $f_\delta(\cdot)$, which is equivalent to the original problem according to Remark 2.2. The classical zeroth-order methods based on such equivalence require to access the gradient estimator to $\nabla f_\delta(\cdot)$ by stochastic function values [6, 32, 35, 36]. Different from the classical methods, we can take the advantage of the quantum estimators, which can be constructed by accessing quantum stochastic function value oracles due to our novel results in Section 3.

We first propose an algorithm which uses the quantum gradient estimator to replace $\nabla f_\delta(\mathbf{x})$ to do the gradient descent step at each iteration. We present the quantum gradient-free method (QGFM) in Algorithm 1. Given a desired variance level $\hat{\sigma}_t^2$, line 2 of Algorithm 1 can be constructed explicitly and efficiently by the quantum stochastic function value oracles $\mathbf{U}_F$ according to Theorem 3.5. The following theorem gives the upper bound on the total $\mathbf{U}_F$ that Algorithm 1 require to access for finding the $(\delta, \epsilon)$-Goldstein stationary point.

**Theorem 4.1.** *Under Assumption 1, by setting the parameter in Algorithm 1 as $\eta = \delta/(2d^{1/2}L)$ and $\hat{\sigma}_t^2 \equiv \epsilon^2/2$, then the total queries of stochastic quantum function value oracle $\mathbf{U}_F$ for finding the $(\delta, \epsilon)$-Goldstein stationary point of $f(\cdot)$ can be bounded by $\tilde{\mathcal{O}}\left(d^{3/2}\left(\frac{L^3}{\epsilon^3} + \frac{L^2\Delta}{\delta\epsilon^3}\right)\right)$, where $\Delta = f(\mathbf{x}_0) - f^*$.*

*Remark* 4.2. QGFM(Algorithm 1) speedups the gradient-free method (GFM) [35] for finding $(\delta, \epsilon)$-stationary point by a factor of $L\epsilon^{-1}$.

In particular, Algorithm 1 utilized a simple gradient descent step to achieve $\Omega(\delta^{-1}\epsilon^{-3})$, which is optimal for classical zeroth-order and first-order methods in terms of $\epsilon$ and $\delta$. It is worth mentioning that the classical methods that achieve this lower bound typically involve multiple loops [6] or rely on additional online optimization algorithms [13, 32].

To further enhance the query complexity in Theorem 4.1, we propose the fast quantum gradient-free method (QGFM+) by incorporating variance reduction techniques, as outlined in Algorithm 2.

---

**Algorithm 2** Fast Quantum Gradient-Free Method (QGFM+)

---

1: Construct $\mathbf{g}_0$ as an unbiased estimator of $\nabla f_\delta(\mathbf{x}_0)$ with variance at most $\hat{\sigma}_{1,0}^2$.

2: **for** $t = 0, 1 \ldots T$

3:      $\mathbf{x}_{t+1} = \mathbf{x}_t - \eta \mathbf{g}_t$

4:      Flip a coin $\theta_t \in \{0, 1\}$ where $P(\theta_t = 1) = p_t$

5:      **If** $\theta_t = 1$ **then**

6:          Construct $\mathbf{g}_{t+1}$ as an unbiased quantum estimator of $\nabla f_\delta(\mathbf{x}_{t+1})$ with variance at most $\hat{\sigma}_{1,t+1}^2$ using $\mathbf{U}_F$ according to Theorem 3.5.

7:      **else**

8:          Construct $\Delta \mathbf{g}_{t+1}$ as an unbiased quantum estimator of $\nabla f_\delta(\mathbf{x}_{t+1}) - \nabla f_\delta(\mathbf{x}_t)$ with variance at most $\hat{\sigma}_{2,t+1}^2$ using $\mathbf{U}_F$ according to Theorem 3.5.

9:          $\mathbf{g}_{t+1} = \mathbf{g}_t + \Delta \mathbf{g}_{t+1}$.

10: **end for**

---

QGFM+ can be seen as a quantum-accelerated version of GFM+ [6]. Unlike GFM+, which required double loops, QGFM+ simplifies the implementation by using a single loop based on the PAGE framework [34]. Moreover, we replace all classical estimators with quantum estimators in lines 6 and 8 of Algorithm 2. These quantum estimators can be constructed efficiently using stochastic quantum function value oracles with a desired variance level, as demonstrated in Theorem 3.5. We present the total number of queries of $\mathbf{U}_F$ for QGFM+ in the following theorem. We present the total queries of $\mathbf{U}_F$ for QGFM+ in the following theorem.

**Theorem 4.3.** *Under Assumption 1, by setting the parameters in Algorithm 2 as follows*

$$\eta = \delta/(2d^{1/2}L), \quad p_t \equiv \epsilon^{2/3}/L^{2/3}, \quad \hat{\sigma}_{1,t}^2 \equiv \epsilon^2/2, \quad and \quad \hat{\sigma}_{2,t}^2 = \epsilon^{2/3}L^{4/3}d\|\mathbf{x}_t - \mathbf{x}_{t-1}\|^2/\delta^2,$$

*then the total queries of stochastic quantum function value oracle $\mathbf{U}_F$ for finding the $(\delta, \epsilon)$-Goldstein stationary point of $f(\cdot)$ can be bounded by $\tilde{\mathcal{O}}\left(d^{3/2}\left(\frac{L^{7/3}}{\epsilon^{7/3}} + \frac{L^{4/3}\Delta}{\delta\epsilon^{7/3}}\right)\right)$, where $\Delta = f(\mathbf{x}_0) - f^*$.*

*Remark* 4.4. QGFM+ (Algorithm 2) speedups the GFM+ [6] for finding $(\delta, \epsilon)$-stationary point by a factor of $L\epsilon^{-2/3}$.

We can see that QGFM+ achieves the query complexity of $\tilde{\mathcal{O}}(d^{3/2}\epsilon^{-7/3}\delta^{-1})$, which cannot be achieved by any of the classical methods. Furthermore, we observe the applicability of our framework to *smooth non-convex* optimization.

*Remark* 4.5. QGFM+ is different from the quantum speedups algorithm (Q-SPIDER) for *non-convex smooth* stochastic optimization [48]: QGFM+ adjusts the variance level of $\Delta \mathbf{g}_t$ according to the difference between the current iteration point and the previous one, while Q-SPIDER fixes the variance levels. Using the adaptive variance level and the QGFM + framework, we can further accelerate the Q-SPIDER for *smooth non-convex* optimization. In Appendix G, we propose the fast quantum gradient method (QGM +) with the query complexity of $\tilde{\mathcal{O}}(\sqrt{d}\epsilon^{-7/3})$, which improves the one of $\tilde{\mathcal{O}}(\sqrt{d}\epsilon^{-5/2})$ obtained in Sidford and Zhang [48].

## 5 Conclusion and Future Work

In this paper, we have presented quantum algorithms for finding the $(\delta, \epsilon)$-Goldstein stationary point for a non-smooth non-convex objective. Our query complexities demonstrate a clearly quantum speedup over the classical methods. In future work, it would be intriguing to explore the framework without ideal distributions which is caused by the limitation of classical or quantum resources. It is also interesting to find the quantum speedups for deterministic methods [14, 28, 51] or the NS-NC objective with constraints [38]. We are also interested in seeing if similar strategies can be applied to quantum online optimization with zeroth-order feedback [25, 26, 33, 56, 58]. The query complexity of the proposed methods still have heavy dependency on the dimension; it is also possible to reduce the dimension dependency based on other quantum techniques and design efficient first-order quantum methods.

## Acknowledgement

Chengchang Liu thanks Luo Luo and Zongqi Wan for a valuable discussion. The work of John C.S. Lui was supported in part by the RGC GRF:14207721.

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

## A Explicit Construction of Quantum Sampling Oracles

In this section, we propose a novel quantum process to realize quantum sampling oracle $\mathbf{O}_{\mathbf{w},\xi}$ : $|\mathbf{0}\rangle \longmapsto \sum_{\mathbf{w},\xi} \sqrt{\Pr[\mathbf{w},\xi]}|\mathbf{w},\xi\rangle|\psi_{\mathbf{w},\xi}\rangle$ with uniform distribution, where $\xi$ is uniformly distributed on $\{0,\cdots,N-1\}$ and $\mathbf{w}$ is sampled uniformly on a discrete unit sphere.

The uniform distribution of $\xi$ in the quantum state can be constructed using Hadamard gates. The construction of a uniform distribution on a discrete unit sphere is more tricky. Classically, such a distribution can be constructed by sampling each coordinate from a standard Gaussian distribution and then normalizing the vector to have unit length by dividing by its norm. However, preparing a superposition state with Gaussian amplitudes is not trivial because the Gaussian distribution is defined in an infinite interval. Constructing such a state with Grover's method [24] will lead to some issues in dealing with the domain and normalization of the measurement probability. Instead, here, starting with the simple uniform superposition state, we use a central limit theorem to construct the standard Gaussian distribution.

The overall quantum algorithm proceeds as follows:

Step 1. Prepare the initial quantum state $|0\rangle^{\otimes m_1} \otimes |0\rangle^{\otimes(dm_2)} \otimes |0\rangle^{\otimes(d\log m_2)}$. Set $k = 0$. Apply $H^{\otimes m_1} \otimes H^{\otimes dm_2} \otimes I$, that is, apply Hadamard gates to the first and second registers. Here, $m_1, m_2 \in \mathbb{N}_+$.

Step 2. Define $h : \{0,1\}^{m_2} \to \mathbb{R}$, $h(\mathbf{j}) = 2\sqrt{m_2}\left(\frac{j_1+j_2+\cdots+j_{m_2}}{\sqrt{m_2}} - 0.5\right)$. Apply $I \otimes U_h^{\otimes d}$, where $U_h$, the unitary transform corresponding to $h$, maps the quantum state $|\mathbf{j}\rangle|0\rangle$ to the quantum state $|\mathbf{j}\rangle|0 + h(\mathbf{j})\rangle$. The $k$-th $U_h$ takes the $k$-th $m_2$ qubits in the second register as input, and the output is stored in the $k$-th $\log m_2$ qubits in the third register, for all $k \in \{0, \ldots, d-1\}$.

Step 3. Consider the third register as a $d$-dimension vector $\mathbf{w}'$, with $\log m_2$ qubits to store each coordinate $\mathbf{w}'_k$. Apply $U_{\text{norm}} : |\mathbf{w}\rangle|0 + \|\mathbf{w}\|\rangle$, the result is stored in an additional ancillary register. Then normalize $\mathbf{w}'$ to have unit length by dividing by $\|\mathbf{w}\|$ in each component.

**Analysis and Correctness.** In Step 1, it starts with the quantum state $|0\rangle^{\otimes m_1} \otimes |0\rangle^{\otimes(dm_2)} \otimes |0\rangle^{\otimes(d\log m_2)}$, where all the registers are initialized to 0. The first register is prepared to create the superposition of $\xi$, and the second and third registers are prepared for creating the superposition of $\mathbf{w}$. We apply Hadamard gates to the first and the second registers, to obtain a uniform superposition of computation basis, which gives

$$\frac{1}{\sqrt{2^{m_1}dm_2}} \sum_{i=0}^{2^{m_1}-1} |i\rangle \otimes \sum_{j_1^{(0)},\ldots,j_{m_2}^{(0)}=0}^{1} \left|j_1^{(0)}\ldots j_{m_2}^{(0)}\right\rangle \otimes \cdots \otimes \sum_{j_1^{(d-1)},\ldots,j_{m_2}^{(d-1)}=0}^{1} \left|j_1^{(d-1)}\ldots j_{m_2}^{(d-1)}\right\rangle \otimes |\mathbf{0}\rangle.$$

Let $m_1 = \lceil \log N \rceil$, and we relabel the first register to obtain

$$\frac{1}{\sqrt{2^{m_1}dm_2}} \sum_{\xi} |\xi\rangle \otimes \sum_{j_1^{(0)},\ldots,j_{m_2}^{(0)}=0}^{1} \left|j_1^{(0)}\ldots j_{m_2}^{(0)}\right\rangle \otimes \cdots \otimes \sum_{j_1^{(d-1)},\ldots,j_{m_2}^{(d-1)}=0}^{1} \left|j_1^{(d-1)}\ldots j_{m_2}^{(d-1)}\right\rangle \otimes |\mathbf{0}\rangle.$$

After Step 2, as each $U_h$ operates in the same manner, we take one as an example,

$$\frac{1}{\sqrt{2^{m_1}dm_2}} \sum_{\xi} |\xi\rangle \otimes \cdots \sum_{j_1,\ldots,j_{m_2}=0}^{1} |j_1 j_2 \ldots j_{m_2}\rangle \ldots \left|2\sqrt{m_2}\left(\frac{j_1+j_2+\cdots+j_{m_2}}{\sqrt{m_2}} - 0.5\right)\right\rangle \ldots.$$

Once measured, $j_1, \ldots, j_{m_2}$ are independent and identically distributed random variables with mean 0.5 and variance 0.25. By the central limit theorem, the average of $\{j_i\}_{i=1}^{m_2}$ approximates the Gaussian distribution when $m_2$ is large. We obtain the standard Gaussian distribution after changing and scaling the average of them. We denote $\mathbf{w}'_k \triangleq 2\sqrt{m_2}\left(\frac{j_1^{(k)}+j_2^{(k)}+\cdots+j_{m_2}^{(k)}}{\sqrt{m_2}} - 0.5\right)$, then the measurement results of $\sum_{\mathbf{j}^{(k)}} |\mathbf{w}'_k\rangle$ follow the distribution of $\mathcal{N}(0,1)$.

After Step 3, the vector in the third register is mapped to the unit sphere and the measurement result follows the uniform distribution on a discrete unit sphere. Rearrange the order of the registers,

denote all the garbage qubits as $|\psi_{\mathbf{w},\xi}\rangle$, we obtain

$$\sum_{\mathbf{w},\xi} \sqrt{\Pr[\mathbf{w},\xi]}\,|\mathbf{w},\xi\rangle\,|\psi_{\mathbf{w},\xi}\rangle,$$

where $\mathbf{w} = \mathbf{w}'/\|\mathbf{w}'\|$ is uniformly distributed on a discrete unit sphere and $\xi$ is uniformly distributed on $\{0,\cdots,N-1\}$.

This realizes the discrete version of the quantum sample oracle $\mathbf{O}_{\mathbf{w},\xi}$ with uniform distribution.

*Remark* A.1. In the ideal scenario where we do not need to limit the number of qubits, allowing $m_1$ and $m_2$ to be sufficiently large, we can achieve $\int_{\mathbf{w}\in S^{d-1}} \sqrt{\mu(\mathbf{w})d\mathbf{w}}|\mathbf{w}\rangle$ as needed in Proposition 3.1. Specifically, our process requires $m_1 + m_2 \times d$ Hadamard gates, $\mathcal{O}(dm_2)$ fundamental arithmetic operations, and 1 calls to the norm circuit. Here, $m_1 = \lceil \log N \rceil$ and $m_2$ is the number of random variables that are used to approximate the Gaussian distribution. Note that many gates here can be performed in parallel, for example, all the $H$ gates can be performed simultaneously, and the sum of $m_2$ qubits can be implemented in a circuit of $\mathcal{O}(\log m_2)$ depth. The total depth complexity is $\mathcal{O}(\log m_2 + \log(d\log m_2))$, which indicates that the depth of the circuit will remain small when $m_2$ increases. This ensures that our construction is feasible even in the context of the NISQ quantum computer [3, 44], which only supports low-depth circuits. In particular, this procedure does not require querying $\mathbf{U}_F$, thus not increasing the query complexity of $\mathbf{U}_F$. Nevertheless, it is still important to give such an explicit and efficient construction to ensure that the quantum state preparation will not ruin the quantum advantage for the overall time complexity.

*Remark* A.2. If $\xi$ is sampled from distribution other than uniform distribution, there still exist quantum techniques which can construct such quantum sample oracle. When detailed classical sampling circuits are known, we can make it reversible by replacing gates in the classical circuits with reversible quantum gates such as the Toffoli gate [42], to obtain a quantum circuit [57]. When there is only a black box access to the classical circuit, we discuss the construction by cases. For the continuous case where the distribution is described by a probability density function, we can use the Grover's method [24], which requires an efficient integrating circuit. For the discrete case, we can extend the Grover's method by using the QRAM data structure. The complexity of constructing a QRAM data structure is linearly dependent on the size of the sample space. Once it is constructed, the complexity of generating the quantum sample oracle depends only logarithmly on the size of sample space [30].

## B  The Proof of Lemma 3.2

*Proof.* First, we claim that a unitary operator $\mathbf{U}_{\mathbf{g},\delta}$ for computing the stochastic gradient estimator $\mathbf{g}_\delta(\cdot;\mathbf{w},\xi)$ can be efficiently constructed. More precisely, we can construct

$$\mathbf{U}_{\mathbf{g},\delta} : |\mathbf{x}\rangle \otimes |\xi\rangle \otimes |\mathbf{w}\rangle \otimes |b\rangle \longmapsto |\mathbf{x}\rangle \otimes |\mathbf{g}_\delta(\mathbf{x};\mathbf{w},\xi)\rangle \otimes |\psi_{\mathbf{w},\xi}\rangle \tag{5}$$

with 2 queries to $\mathbf{U}_F$. Now we assume the access to $\mathbf{U}_{\mathbf{g},\delta}$ and the description of its construction will be deferred to the end of this proof. Next we show how this can lead to a quantum $\delta$-estimated stochastic gradient oracle $\mathbf{O}_{\mathbf{g},\delta}$ as defined in Definition 3.3. Given initial state $|\mathbf{x}\rangle \otimes |\mathbf{0}\rangle \otimes |\mathbf{0}\rangle$, we can prepare the desired quantum state by first applying the quantum sampling oracle $\mathbf{O}_{\mathbf{w},\xi}$ and then $\mathbf{U}_{\mathbf{g},\delta}$ as follows:

$$\begin{aligned}
\mathbf{U}_{\mathbf{g},\delta} \cdot (\mathbf{I} \otimes \mathbf{O}_{\xi,\mathbf{w}} \otimes \mathbf{I})|\mathbf{x}\rangle \otimes |\mathbf{0}\rangle \otimes |\mathbf{0}\rangle &= \mathbf{U}_{\mathbf{g},\delta}\big(|\mathbf{x}\rangle \otimes \sum_{\xi,\mathbf{w}} \sqrt{p(\xi,\mathbf{w})}|\xi,\mathbf{w}\rangle \otimes |\mathbf{0}\rangle\big)\\
&= \sum_{\xi,\mathbf{w}} \sqrt{p(\xi,\mathbf{w})}\,\mathbf{U}_{\mathbf{g},\delta}\big(|\mathbf{x}\rangle \otimes |\xi,\mathbf{w}\rangle \otimes |\mathbf{0}\rangle\big)\\
&= |\mathbf{x}\rangle \otimes \sum_{\xi,\mathbf{w}} \sqrt{p(\xi,\mathbf{w})}|\mathbf{g}_\delta(\mathbf{x};\mathbf{w},\xi)\rangle \otimes |\psi_{\mathbf{w},\xi}\rangle.
\end{aligned}$$

Next we finish the proof by presenting how to implement $\mathbf{U}_{\mathbf{g},\delta}$ with two queries to $\mathbf{U}_F$. Since $\delta$ and $d$ are fixed and known beforehand, we can easily construct the following three operators via the quantum unitary implementations of the corresponding classical arithmetic operations:

$$\mathbf{A}_+ : |\mathbf{x}\rangle \otimes |\mathbf{w}\rangle \otimes |\mathbf{0}\rangle \longmapsto |\mathbf{x}\rangle \otimes |\mathbf{w}\rangle \otimes |\mathbf{x}+\delta\mathbf{w}\rangle, \quad \mathbf{A}_- : |\mathbf{x}\rangle \otimes |\mathbf{w}\rangle \otimes |\mathbf{0}\rangle \longmapsto |\mathbf{x}\rangle \otimes |\mathbf{w}\rangle \otimes |\mathbf{x}-\delta\mathbf{w}\rangle,$$

$$\mathsf{sub} : |a\rangle \otimes |b\rangle \longmapsto |a\rangle \otimes |a - b\rangle, \qquad \text{and} \qquad \mathsf{Fmul} : |c\rangle \longmapsto \left|\frac{\delta}{2d}c\right\rangle.$$

Let $F'(\mathbf{x}; \mathbf{w}, \xi) \triangleq \frac{\delta}{2d}\left(F(\mathbf{x} + \delta\mathbf{w}; \xi) - F(\mathbf{x} - \delta\mathbf{w}; \xi)\right)$. Then we construct a unitary $\mathbf{D}$ as follows:

$$
\begin{aligned}
\mathbf{D} : \quad & |\mathbf{x}\rangle \otimes |\xi\rangle \otimes |\mathbf{w}\rangle \otimes |\mathbf{0}\rangle \otimes |\mathbf{0}\rangle \otimes |\mathbf{0}\rangle \otimes |\mathbf{0}\rangle \\
&\mapsto_{(a)} |\mathbf{x}\rangle \otimes |\xi\rangle \otimes |\mathbf{w}\rangle \otimes |\mathbf{0}\rangle \otimes |\mathbf{0}\rangle \otimes |\mathbf{x} - \delta\mathbf{w}\rangle \otimes |\mathbf{0}\rangle \\
&\mapsto_{(b)} |\mathbf{x}\rangle \otimes |\xi\rangle \otimes |\mathbf{w}\rangle \otimes |F(\mathbf{x} - \delta\mathbf{w}; \xi)\rangle \otimes |\mathbf{0}\rangle \otimes |\mathbf{x} - \delta\mathbf{w}\rangle \otimes |\mathbf{0}\rangle \\
&\mapsto_{(c)} |\mathbf{x}\rangle \otimes |\xi\rangle \otimes |\mathbf{w}\rangle \otimes |F(\mathbf{x} - \delta\mathbf{w}; \xi)\rangle \otimes |F(\mathbf{x} + \delta\mathbf{w}; \xi)\rangle \otimes |\mathbf{x} - \delta\mathbf{w}\rangle \otimes |\mathbf{x} + \delta\mathbf{w}\rangle \\
&\mapsto_{(d)} |\mathbf{x}\rangle \otimes |\xi\rangle \otimes |\mathbf{w}\rangle \otimes |F'(\mathbf{x}; \mathbf{w}, \xi)\rangle \otimes |F(\mathbf{x} + \delta\mathbf{w}; \xi)\rangle \otimes |\mathbf{x} - \delta\mathbf{w}\rangle \otimes |\mathbf{x} + \delta\mathbf{w}\rangle \\
&= |\mathbf{x}\rangle \otimes |\xi\rangle \otimes |\mathbf{w}\rangle \otimes |F'(\mathbf{x}; \mathbf{w}, \xi)\rangle \otimes |\psi'_{\mathbf{w}, \xi}\rangle,
\end{aligned}
\tag{6}
$$

where $(a)$ follows by applying $\mathbf{A}_-$ on the first, third and sixth registers; $(b)$ uses the quantum stochastic function value oracle $\mathbf{U}_F$ on the second, fourth and sixth registers; $(c)$ uses $\mathbf{A}_+$ and $\mathbf{U}_F$ in a way similar to steps $(a)$ and $(b)$; $(d)$ applies sub on the fourth and fifth registers, and then applies Fmul on the fourth register. It is easy to see that this unitary $\mathbf{D}$ uses only 2 queries to $\mathbf{U}_F$.

For any input state $|\mathbf{x}\rangle \otimes |\mathbf{w}, \xi\rangle \otimes |0\rangle \otimes |0\rangle^{\otimes d}$, apply $\mathbf{D}$ to obtain

$$
\begin{aligned}
& |\mathbf{x}\rangle \otimes |\xi\rangle \otimes |\mathbf{w}\rangle \otimes |F'(\mathbf{x}; \mathbf{w}, \xi)\rangle \otimes |\psi'_{\mathbf{w}, \xi}\rangle \otimes |0\rangle^{\otimes d} \\
&= |\mathbf{x}\rangle \otimes |\xi\rangle \otimes |w_1, \cdots, w_d\rangle \otimes |F'(\mathbf{x}; \mathbf{w}, \xi)\rangle \otimes |\psi'_{\mathbf{w}, \xi}\rangle \otimes |0\rangle^{\otimes d}
\end{aligned}
\tag{7}
$$

Next we will utilize quantum multiplication operator $\mathbf{U}_{\mathsf{mul}} : |a\rangle \otimes |b\rangle \otimes |c\rangle \longrightarrow |a\rangle \otimes |b\rangle \otimes |c \oplus ab\rangle$. This can be implemented by the quantization of classical multiplication algorithms, whose details can be found in [20, 45, 46].

Applying $\mathbf{U}_{\mathbf{mul}}$ to each $|w_i, F'\rangle \otimes |0\rangle$ for all $i \in [d]$ yields

$$
\begin{aligned}
& |\mathbf{x}\rangle \otimes |\xi\rangle \otimes |w_1, \cdots, w_d\rangle \otimes |F'(\mathbf{x} + \delta\mathbf{w}; \xi)\rangle \otimes |\psi'_{\mathbf{w}, \xi}\rangle \otimes |F'(\mathbf{x} + \delta\mathbf{w}; \xi)w_1, \cdots, F'(\mathbf{x} + \delta\mathbf{w}; \xi)w_d\rangle \\
&= |\mathbf{x}\rangle \otimes |\xi\rangle \otimes |w_1, \cdots, w_d\rangle \otimes |F'(\mathbf{x} + \delta\mathbf{w}; \xi)\rangle \otimes |\psi'_{\mathbf{w}, \xi}\rangle \otimes |F'(\mathbf{x} + \delta\mathbf{w}; \xi)\mathbf{w}\rangle \\
&= |\mathbf{x}\rangle \otimes |\xi\rangle \otimes |w_1, \cdots, w_d\rangle \otimes |F'(\mathbf{x} + \delta\mathbf{w}; \xi)\rangle \otimes |\psi'_{\mathbf{w}, \xi}\rangle \otimes |\mathbf{g}_\delta(\mathbf{x}; \mathbf{w}, \xi)\rangle \\
&= |\mathbf{x}\rangle \otimes |\psi_{\mathbf{w}, \xi}\rangle \otimes |\mathbf{g}_\delta(\mathbf{x}; \mathbf{w}, \xi)\rangle.
\end{aligned}
\tag{8}
$$

By swapping the last two quantum registers, we obtain $|\mathbf{x}\rangle \otimes |\mathbf{g}(\mathbf{x}; \mathbf{w}, \xi)\rangle \otimes |\psi_{\mathbf{w}, \xi}\rangle$. Hence, $\mathbf{U}_{\mathbf{g}, \delta}$ can be implemented with two queries to $\mathbf{U}_F$. $\square$

## C   The Proof of Corollary 3.4

*Proof.* Analogous to eq. (5), we claim that the following unitary $\mathbf{V}_{\mathbf{g}, \delta}$ can be implemented with 4 queries to $\mathbf{U}_F$:

$$\mathbf{V}_{\mathbf{g}, \delta} : |\mathbf{x}\rangle \otimes |\mathbf{y}\rangle \otimes |\xi\rangle \otimes |\mathbf{w}\rangle \otimes |b\rangle \longmapsto |\mathbf{x}\rangle \otimes |\mathbf{y}\rangle \otimes |\mathbf{g}_\delta(\mathbf{x}; \mathbf{w}, \xi) - \mathbf{g}_\delta(\mathbf{y}; \mathbf{w}, \xi)\rangle \otimes |\psi_{\mathbf{w}, \xi}\rangle.$$

With access to $\mathbf{V}_{\mathbf{g}, \delta}$ and $\mathbf{O}_{\mathbf{g}, \delta}$, we can construct $\mathbf{O}_{\Delta\mathbf{g}_\delta}$ as

$$\mathbf{O}_{\Delta\mathbf{g}_\delta} = \mathbf{V}_{\mathbf{g}, \delta} \cdot (\mathbf{I} \otimes \mathbf{I} \otimes \mathbf{O}_{\xi, \mathbf{w}} \otimes \mathbf{I}).$$

Next, to implement $\mathbf{V}_{\mathbf{g}, \delta}$ with 4 queries to $\mathbf{U}_F$, we can first follow the steps in eq. (6), eq. (7) and eq. (8) to get a unitary that performs the mapping below

$$|\mathbf{x}\rangle \otimes |\mathbf{y}\rangle \otimes |\xi\rangle \otimes |\mathbf{w}\rangle \otimes |\mathbf{0}\rangle \otimes |\mathbf{0}\rangle \otimes |\mathbf{0}\rangle \otimes |\mathbf{0}\rangle \longmapsto |\mathbf{x}\rangle \otimes |\mathbf{y}\rangle \otimes |\psi_{\mathbf{w}, \xi}\rangle \otimes |\mathbf{g}_\delta(\mathbf{x}; \mathbf{w}, \xi)\rangle \otimes |\mathbf{g}_\delta(\mathbf{y}; \mathbf{w}, \xi)\rangle.$$

Then applying sub and a SWAP gate to the output above yields

$$|\mathbf{x}\rangle \otimes |\mathbf{y}\rangle \otimes \sum_{\xi, \mathbf{w}} \sqrt{\Pr[\mathbf{w}, \xi]} |\mathbf{g}_\delta(\mathbf{x}; \mathbf{w}, \xi) - \mathbf{g}_\delta(\mathbf{y}; \mathbf{w}, \xi)\rangle \otimes |\psi_{\mathbf{w}, \xi}\rangle.$$

$\square$

# D The Proof of Theorem 3.5

Before we present the proof, we first introduce the results for the quantum mean estimation by Sidford and Zhang [48].

**Theorem D.1** ([48, Theorem 4]). *For a random variable $X$ with bounded variance such that* $\mathrm{Var}[X] \le \hat{L}^2$, *there exists an algorithm that can output an unbiased estimator $\hat{\mu}$ of $\mu = \mathbb{E}[X]$ satisfying* $\mathbb{E}[\|\hat{\mu} - \mu\|^2] \le \hat{\sigma}^2$ *using an expected* $\tilde{\mathcal{O}}(\hat{L}\sqrt{d}\hat{\sigma}^{-1})$ *queries of quantum sampling oracle* $\mathbf{O}_X$ *as defined in Definition 3.2.*

*Proof.* According to Proposition 3.1, the quantum $\delta$-estimated stochastic gradient oracle given the input $\mathbf{x}$ satisfies that

$$\mathbb{E}[\mathbf{g}_\delta] = \nabla f_\delta(\mathbf{x}) \quad \text{and} \quad \mathrm{Var}[\mathbf{g}_\delta] \le 16\sqrt{2}\pi dL^2.$$

Using Theorem D.1 with $\hat{L} = \sqrt{d}L$, it requires only $\tilde{\mathcal{O}}(dL\hat{\sigma}_1^{-1})$ queries of $\mathbf{O}_{\mathbf{g}_\delta}$ to construct the quantum estimator $\hat{\mathbf{g}}$ such that $\mathbb{E}[\|\hat{\mathbf{g}} - \nabla f_\delta(\mathbf{x})\|^2] \le \hat{\sigma}_1^2$. According to Lemma 3.2, we can construct each $\mathbf{O}_{\mathbf{g}_\delta}$ by $\mathcal{O}(1)$-queries of $\mathbf{U}_F$. Thus, it only requires $\tilde{\mathcal{O}}(dL\hat{\sigma}_1^{-1})$ queries of $\mathbf{U}_F$ to construct the mini-batch quantum estimator $\hat{\mathbf{g}}$.

Similarly, since we can construct the quantum estimator $\Delta\mathbf{g}_\delta$ by $\mathcal{O}(1)$-queries of $\mathbf{U}_F$ according to Corollary 3.4, with the following properties

$$\mathbb{E}[\Delta\mathbf{g}_\delta] = \nabla f_\delta(\mathbf{x}) - \nabla f_\delta(\mathbf{y}) \quad \text{and} \quad \mathrm{Var}[\Delta\mathbf{g}_\delta] \le \mathbb{E}[\|\Delta\mathbf{g}_\delta\|^2] \le d^2 L^2 \delta^{-2} \|\mathbf{x} - \mathbf{y}\|^2,$$

then, using Theorem D.1 with $\hat{L} = dL\delta^{-1}\|\mathbf{x} - \mathbf{y}\|$ directly leads to second statement. $\square$

# E The Proof of Theorem 4.1

*Proof.* According to the variance level we set, $\mathbf{g}_t$ satisfies that

$$\mathbb{E}[\|\mathbf{g}_t - \nabla f_\delta(\mathbf{x}_t)\|^2] \le \frac{\epsilon^2}{2}.$$

According to Proposition 2.1, $f_\delta(\cdot)$ is a nonconvex function, with $(\sqrt{d}L\delta^{-1})$-Lipschitz gradient, which implies that

$$f_\delta(\mathbf{x}_{t+1}) \le f_\delta(\mathbf{x}_t) + \langle \nabla f_\delta(\mathbf{x}), \mathbf{x}_{t+1} - \mathbf{x}_t \rangle + \frac{\sqrt{d}L\delta^{-1}}{2}\|\mathbf{x}_{t+1} - \mathbf{x}_t\|^2$$

$$= f_\delta(\mathbf{x}_t) - \eta\langle \nabla f_\delta(\mathbf{x}), \mathbf{g}_t \rangle + \frac{\sqrt{d}L\delta^{-1}}{2}\|\mathbf{x}_{t+1} - \mathbf{x}_t\|^2$$

Taking expectation on both sides of the above inequality, we have

$$f_\delta(\mathbf{x}_{t+1}) \le f_\delta(\mathbf{x}_t) - \eta\|\nabla f_\delta(\mathbf{x}_t)\|^2 + \frac{\eta^2\sqrt{d}L\delta^{-1}}{2}\mathbb{E}[\|\mathbf{g}_t\|^2]$$

$$\le f_\delta(\mathbf{x}_t) - \eta\|\nabla f_\delta(\mathbf{x}_t)\|^2 + \eta^2\sqrt{d}L\delta^{-1}\left(\|\nabla f_\delta(\mathbf{x}_t)\|^2 + \mathbb{E}[\|\mathbf{g}_t - \nabla_\delta(\mathbf{x}_t)\|^2]\right)$$

$$\le f_\delta(\mathbf{x}_t) - \left(\eta - \sqrt{d}L\delta^{-1}\eta^2\right)\|\nabla f_\delta(\mathbf{x}_t)\|^2 + \sqrt{d}L\delta^{-1}\eta^2 \cdot \frac{\epsilon^2}{2},$$

We let $\eta = \frac{\delta}{2\sqrt{d}L}$, then it holds that

$$\mathbb{E}[\|\nabla f_\delta(\mathbf{x}_t)\|^2] \le 2\sqrt{d}L\delta^{-1}\left(f_\delta(\mathbf{x}_t) - f_\delta(\mathbf{x}_{t+1})\right) + \frac{\epsilon^2}{4}.$$

Summing up the above inequality, we have

$$\mathbb{E}\left[\frac{\sum_{t=0}^{T}\|\nabla f_\delta(\mathbf{x}_t)\|^2}{T}\right] \le \frac{2\sqrt{d}L\delta^{-1}(f_\delta(\mathbf{x}_0) - f_\delta^*)}{T} + \frac{\epsilon^2}{4} \le \frac{2\sqrt{d}L\delta^{-1}(f(x_0) - f^* + 2\delta L)}{T} + \frac{\epsilon^2}{4}.$$

By setting

$$T = \left\lceil 2\epsilon^{-2}(4\sqrt{d}L^2 + 2\sqrt{d}L\delta^{-1}\Delta) \right\rceil,$$

and choosing $\mathbf{x}_{\text{out}}$ randomly from $\{\mathbf{x}_0, \cdots, \mathbf{x}_{T-1}\}$, we have

$$\mathbb{E}\left[\|\nabla f_\delta(\mathbf{x}_{\text{out}})\|^2\right] \le \frac{1}{T}\mathbb{E}\left[\sum_{i=1}^{T} \|\nabla f_\delta(\mathbf{x}_t)\|^2\right] \le \frac{\epsilon^2}{4} + \frac{\epsilon^2}{2} \le \epsilon^2.$$

Using Theorem 3.5, we require

$$b = \tilde{\mathcal{O}}(dL\epsilon^{-1}),$$

to achieve the desired variance level. Thus the total quantum query of $\mathbf{U}_F$ can be bounded by

$$b \cdot T = \tilde{\mathcal{O}}\left(d^{3/2}\left(\frac{L\Delta}{\epsilon^3\delta} + \frac{L^2}{\epsilon^3}\right)\right).$$

$\square$

# F  The Proof of Theorem 4.3

*Proof.* We denote $L_\delta \triangleq \frac{\sqrt{d}L}{\delta}$. We also denote $\hat{\mathbf{g}}_{t+1}$ as the unbiased estimator of $\nabla f_\delta(\mathbf{x}_{t+1})$ we have constructed in line 6 and $\Delta\mathbf{g}_{t+1}$ as the unbiased estimator of $\nabla f_\delta(\mathbf{x}_{t+1}) - \nabla f_\delta(\mathbf{x}_t)$ we have constructed in line 8. We can see that $\mathbf{g}_{t+1}$ is equivalent to

$$\mathbf{g}_{t+1} = \begin{cases} \hat{\mathbf{g}}_{t+1} & \text{with probability } p_t \\ \mathbf{g}_t + \Delta\mathbf{g}_{t+1} & \text{with probability } 1 - p_t \end{cases}.$$

According to the variance level we set in Theorem 4.3, we have

$$\mathbb{E}\left[\|\hat{\mathbf{g}}_{t+1} - \nabla f_\delta(\mathbf{x}_{t+1})\|^2\right] \le \hat{\sigma}_{1,t+1}^2 = \frac{\epsilon^2}{2},$$

and

$$\mathbb{E}\left[\|\Delta\mathbf{g}_{t+1} - (\nabla f_\delta(\mathbf{x}_{t+1}) - \nabla f_\delta(\mathbf{x}_t))\|^2\right] \le \hat{\sigma}_{2,t+1}^2 = \epsilon^{2/3}\|\mathbf{x}_{t+1} - \mathbf{x}_t\|^2\frac{L^{4/3}d}{\delta^2}.$$

According to Proposition 2.1, $\nabla f_\delta(\cdot)$ is $L_\delta$-Lipschitz continuous, which means

$$
\begin{aligned}
f_\delta(\mathbf{x}_{t+1}) &\le f_\delta(\mathbf{x}_t) + \langle \nabla f_\delta(\mathbf{x}_t), \mathbf{x}_{t+1} - \mathbf{x}_t \rangle + \frac{L_\delta}{2}\|\mathbf{x}_{t+1} - \mathbf{x}_t\|^2 \\
&= f_\delta(\mathbf{x}_t) + \langle \nabla f_\delta(\mathbf{x}_t) - \mathbf{g}_t, \mathbf{x}_{t+1} - \mathbf{x}_t \rangle + \langle \mathbf{g}_t, \mathbf{x}_{t+1} - \mathbf{x}_t \rangle + \frac{L_\delta}{2}\|\mathbf{x}_{t+1} - \mathbf{x}_t\|^2 \qquad (9) \\
&\le f_\delta(\mathbf{x}_t) - \frac{\eta}{2}\|\nabla f_\delta(\mathbf{x}_t)\|^2 - \frac{\eta}{2}\|\mathbf{g}_t - \nabla f_\delta(\mathbf{x}_t)\|^2 - \left(\frac{1}{2\eta} - \frac{L_\delta}{2}\right)\|\mathbf{x}_{t+1} - \mathbf{x}_t\|^2.
\end{aligned}
$$

On the other hand, we track the variance of $\mathbf{g}_{t+1}$ by

$$
\begin{aligned}
&\mathbb{E}\left[\|\mathbf{g}_{t+1} - \nabla f_\delta(\mathbf{x}_{t+1})\|^2\right] \\
&= p_t\mathbb{E}\left[\|\hat{\mathbf{g}}_{t+1} - \nabla f_\delta(\mathbf{x}_{t+1})\|^2\right] \\
&\quad + (1-p_t)\mathbb{E}\left[\|\mathbf{g}_t - \nabla f_\delta(\mathbf{x}_t) + (\Delta\mathbf{g}_{t+1} - (\nabla f_\delta(\mathbf{x}_{t+1}) - \nabla f_\delta(\mathbf{x}_t)))\|^2\right] \qquad (10) \\
&= p_t\epsilon^2 + (1-p_t)\|\mathbf{g}_t - \nabla f_\delta(\mathbf{x}_t)\|^2 + (1-p_t) \cdot \frac{L^{4/3}\epsilon^{2/3}d}{\delta^2}\|\mathbf{x}_{t+1} - \mathbf{x}_t\|^2.
\end{aligned}
$$

We have $p_t \equiv p$ and can denote $\Phi_t \triangleq f_\delta(\mathbf{x}_t) - f^* + \frac{\eta}{2p}\|\mathbf{g}_t - \nabla f_\delta(\mathbf{x}_t)\|^2$. Combining eq. (9) and eq. (10), we have

$$
\begin{aligned}
\mathbb{E}\left[\Phi_{t+1}\right] &= \mathbb{E}\left[f_\delta(\mathbf{x}_{t+1}) + \frac{\eta}{2p}\|\mathbf{g}_{t+1} - \nabla f(\mathbf{x}_{t+1})\|^2\right] \\
&\le \mathbb{E}\left[f_\delta(\mathbf{x}_t) - \frac{\eta}{2}\|\nabla f_\delta(\mathbf{x}_t)\|^2 - \frac{\eta}{2}\|\mathbf{g}_t - \nabla f_\delta(\mathbf{x}_t)\|^2 - \left(\frac{1}{2\eta} - \frac{L_\delta}{2}\right)\|\mathbf{x}_{t+1} - \mathbf{x}_t\|^2\right] \\
&\quad + \frac{\eta}{2p}\mathbb{E}\left[p\epsilon^2 + (1-p)\|\mathbf{g}_t - \nabla f_\delta(\mathbf{x}_t)\|^2 + (1-p)\frac{L^{4/3}d\epsilon^{2/3}}{\delta^2}\|\mathbf{x}_{t+1} - \mathbf{x}_t\|^2\right] \\
&\le \mathbb{E}\left[\Phi_t\right] - \frac{\eta}{2}\|\nabla f_\delta(\mathbf{x}_t)\|^2 - \underbrace{\left(\frac{1}{2\eta} - \frac{L_\delta}{2} - \frac{\eta(1-p)}{p}\cdot\left(\frac{L^{4/3}d\epsilon^{2/3}}{\delta^2}\right)\right)}_{A}\|\mathbf{x}_{t+1} - \mathbf{x}_t\|^2 + \frac{\eta\epsilon^2}{2}.
\end{aligned}
$$

(11)

We have chosen

$$
\eta = \frac{1}{2L_\delta} \quad \text{and} \quad p = \frac{\epsilon^{2/3}}{L^{2/3}},
$$

(12)

such that

$$
A \ge \frac{\sqrt{d}L}{2\delta} - \frac{\delta}{2\sqrt{d}L}\cdot\frac{L^{2/3}}{\epsilon^{2/3}}\cdot\frac{L^{4/3}d\epsilon^{2/3}}{\delta^2} = 0.
$$

Then, eq. (11) implies

$$
\mathbb{E}\left[\|\nabla f_\delta(\mathbf{x}_t)\|^2\right] \le \frac{2}{\eta}\mathbb{E}\left[\Phi_t - \Phi_{t+1}\right] + \epsilon^2.
$$

(13)

Since it holds that

$$
\begin{aligned}
\frac{2}{\eta}\mathbb{E}\left[\Phi_0 - \Phi_T\right] &\le \frac{2}{\eta}\mathbb{E}\left[f_\delta(\mathbf{x}_0) - f_\delta^* + \frac{\eta}{2p}\|\hat{\mathbf{g}}_0 - \nabla f(\mathbf{x}_0)\|^2\right] \\
&\le \frac{2}{\eta}\mathbb{E}\left[f(\mathbf{x}_0) - f^* + 2\delta L + \frac{\eta}{2p}\|\hat{\mathbf{g}}_0 - \nabla f(\mathbf{x}_0)\|^2\right] \\
&\le \frac{2}{\eta}\left(\Delta + 2\delta L\right) + \frac{1}{p}\epsilon^2,
\end{aligned}
$$

summing up eq. (13) from $t = 0, \cdots, T-1$, we have

$$
\frac{1}{T}\sum_{i=0}^{T-1}\mathbb{E}\left[\|\nabla f_\delta(\mathbf{x}_i)\|^2\right] \le \frac{2}{\eta T}\mathbb{E}\left[\Phi_0 - \Phi_T\right] + \frac{\epsilon^2}{2}.
$$

By choosing

$$
T = \left\lceil 8L_\delta\epsilon^{-2}\left(\Delta + 2\delta L\right) + \frac{4}{p}\right\rceil,
$$

(14)

we have

$$
\mathbb{E}\left[\|\nabla f_\delta(\mathbf{x}_{\text{out}})\|^2\right] = \frac{1}{T}\sum_{i=0}^{T-1}\mathbb{E}\left[\|\nabla f_\delta(\mathbf{x}_i)\|^2\right] \le \frac{2}{\eta T}\mathbb{E}\left[\Phi_0 - \Phi_T\right] + \frac{\epsilon^2}{2} \le \frac{\epsilon^2}{4} + \frac{\epsilon^2}{4} + \frac{\epsilon^2}{2} = \epsilon^2.
$$

Using Theorem 3.5, the expectation queries of $\mathbf{U}_F$ to construct $\hat{\mathbf{g}}_t$ is

$$
b_0 = \tilde{\mathcal{O}}\left(dL\hat{\sigma}_{1,t}^{-1}\right) = \tilde{\mathcal{O}}\left(dL\epsilon^{-1}\right),
$$

and the expectation queries of $\mathbf{U}_F$ to construct $\Delta\mathbf{g}_t$ is

$$
b_1 = \tilde{\mathcal{O}}\left(d^{3/2}L\|\mathbf{x}_{t-1} - \mathbf{x}_t\|\hat{\sigma}_{2,t}^{-1}\delta^{-1}\right) = \tilde{\mathcal{O}}\left(dL^{1/3}\epsilon^{-1/3}\right).
$$

Thus, the total quantum queries of $\mathbf{U}_F$ for finding the $(\delta, \epsilon)$-stationary point of $f(\cdot)$ can be bounded by

$$
\begin{aligned}
\tilde{\mathcal{O}}(T(b_0 p + b_1(1-p))) &= \tilde{\mathcal{O}}\left(\sqrt{d}L\epsilon^2(\Delta + 2\delta L)\cdot\left(dL\epsilon^{-1}L^{-2/3}\epsilon^{2/3} + dL^{1/3}\epsilon^{-1/3}\right)\right) \\
&= \tilde{\mathcal{O}}\left(d^{3/2}\left(\frac{L^{4/3}\Delta}{\epsilon^{7/3}\delta} + \frac{L^{7/3}}{\epsilon^{7/3}}\right)\right),
\end{aligned}
$$

which finishes the proof.

$\square$

**Algorithm 3** Fast Quantum Gradient Method (QGM+)

---

1: Construct $\mathbf{g}_0$ as an unbiased estimator of $\nabla f(\mathbf{x}_0)$ with variance at most $\hat{\sigma}_{1,0}^2$.

2: **for** $t = 0, 1 \ldots T$

3:      $\mathbf{x}_{t+1} = \mathbf{x}_t - \eta\mathbf{g}_t$

4:      Flip a coin $\theta_t \in \{0, 1\}$ where $P(\theta_t = 1) = p_t$

5:      **If** $\theta_t = 1$ **then**

6:          Construct $\mathbf{g}_{t+1}$ as an unbiased quantum estimator of $\nabla f(\mathbf{x}_{t+1})$ with variance at most $\hat{\sigma}_{1,t+1}^2$.

7:      **else**

8:          Construct $\Delta\mathbf{g}_{t+1}$ as an unbiased quantum estimator of $\nabla f(\mathbf{x}_{t+1}) - \nabla f(\mathbf{x}_t)$ with variance at most $\hat{\sigma}_{2,t+1}^2$.

9:          $\mathbf{g}_{t+1} = \mathbf{g}_t + \Delta\mathbf{g}_{t+1}$.

10: **end for**

---

# G   Improved Results for Quantum Stochastic Smooth Non-convex Optimization

Sidford and Zhang [48] introduced Q-SPIDER for *smooth non-convex* optimization, with the query complexity of $\tilde{\mathcal{O}}(d^{1/2}\epsilon^{-5/2})$ in the quantum stochastic gradient oracle. Using the same framework as QGFM+, we propose the fast quantum gradient method (QGM+), which further improves the query complexity of Q-SPIDER.

We present QGM+ in Algorithm 3. The main difference between QGFM+ and QGM+ is that QGM constructs estimators for $\nabla f(\mathbf{x})$ and $\nabla f(\mathbf{x}) - \nabla f(\mathbf{y})$ instead of their smoothed surrogates in line 6 and line 8 by using the quantum stochastic gradient oracle directly [48, Definition 4]. We present the setting for Q-SPIDER as follows as being self-contained.

**Assumption 2** ([48, Setting of Theorem 7]). *We assume that we are able to access the quantum stochastic oracle that outputs $\nabla F(\cdot; \xi)$ which is a stochastic gradient of $f(\cdot)$ that satisfies*

$$\mathbb{E}_\xi[\nabla F(\mathbf{x}; \xi)] = \nabla f(\mathbf{x}), \quad \mathbb{E}_\xi\left[\|\nabla F(\mathbf{x}; \xi) - \nabla f(\mathbf{x})\|\right] \le \sigma^2,$$

*and*

$$\mathbb{E}_\xi\left[\|\nabla F(\mathbf{x}; \xi) - \nabla F(\mathbf{y}; \xi)\|^2\right] \le l^2\|\mathbf{x} - \mathbf{y}\|^2.$$

We also present the definition of the $\epsilon$-stationary point of a smooth function.

**Definition G.1.** *We say $\mathbf{x}$ is an $\epsilon$-stationary point of a smooth function $f(\cdot)$, if it satisfies $\|\nabla f(\mathbf{x})\| \le \epsilon$.*

We present the query complexity of QGM+ in the following theorem.

**Theorem G.1.** *Under the same setting of [48, Theorem 7] for Q-SPIDER, QGM+ (Algorithm 3) finds the $\epsilon$-stationary point of $f(\cdot)$ using an expected $\tilde{\mathcal{O}}(\sqrt{d}\epsilon^{-7/3})$ queries of quantum stochastic gradient oracle by setting*

$$\eta = \frac{1}{2l}, \quad p_t \equiv \epsilon^{2/3}\sigma^{-2/3}, \quad \hat{\sigma}_{1,t}^2 \equiv \frac{\epsilon^2}{2}, \quad and \quad \hat{\sigma}_{2,t}^2 = \frac{l^2\epsilon^{2/3}\|\mathbf{x}_t - \mathbf{x}_{t-1}\|}{\sigma^{2/3}}.$$

*Proof.* According to the variance level we set in Theorem G.1 We have

$$\mathbb{E}\left[\|\hat{\mathbf{g}}_{t+1} - \nabla f(\mathbf{x}_{t+1})\|^2\right] \le \hat{\sigma}_{1,t+1}^2 = \frac{\epsilon^2}{2},$$

and

$$\mathbb{E}\left[\|\Delta\mathbf{g}_{t+1} - (\nabla f(\mathbf{x}_{t+1}) - \nabla f(\mathbf{x}_t))\|^2\right] \le \hat{\sigma}_{2,t+1}^2 = \frac{l^2\epsilon^{2/3}}{\sigma^{2/3}}\|\mathbf{x} - \mathbf{y}\|^2$$

$$f(\mathbf{x}_{t+1}) \le f(\mathbf{x}_t) + \langle \nabla f(\mathbf{x}_t), \mathbf{x}_{t+1} - \mathbf{x}_t \rangle + \frac{l}{2}\|\mathbf{x}_{t+1} - \mathbf{x}_t\|^2$$

$$= f(\mathbf{x}_t) + \langle \nabla f(\mathbf{x}_t) - \mathbf{g}_t, \mathbf{x}_{t+1} - \mathbf{x}_t \rangle + \langle \mathbf{g}_t, \mathbf{x}_{t+1} - \mathbf{x}_t \rangle + \frac{l}{2}\|\mathbf{x}_{t+1} - \mathbf{x}_t\|^2 \qquad (15)$$

$$\le f(\mathbf{x}_t) - \frac{\eta}{2}\|\nabla f(\mathbf{x}_t)\|^2 - \frac{\eta}{2}\|\mathbf{g}_t - \nabla f(\mathbf{x}_t)\|^2 - \left(\frac{1}{2\eta} - \frac{l}{2}\right)\|\mathbf{x}_{t+1} - \mathbf{x}_t\|^2.$$

The variance of $\mathbf{g}_{t+1}$ can be traced by

$$\mathbb{E}\left[\|\mathbf{g}_{t+1} - \nabla f(\mathbf{x}_{t+1})\|^2\right]$$

$$= p_t \mathbb{E}\left[\|\hat{\mathbf{g}}_{t+1} - \nabla f(\mathbf{x}_{t+1})\|^2\right]$$

$$+ (1 - p_t)\mathbb{E}\left[\|\mathbf{g}_t - \nabla f(\mathbf{x}_t) + (\Delta \mathbf{g}_{t+1} - (\nabla f(\mathbf{x}_{t+1}) - \nabla f(\mathbf{x}_t)))\|^2\right] \qquad (16)$$

$$= p_t \epsilon^2 + (1 - p_t)\|\mathbf{g}_t - \nabla f(\mathbf{x}_t)\|^2 + (1 - p_t)\frac{l^2 \epsilon^{2/3}}{\sigma^{2/3}} \cdot \|\mathbf{x}_{t+1} - \mathbf{x}_t\|^2.$$

We let $p_t \equiv p$ and denote $\Phi_t \triangleq f(\mathbf{x}_t) - f^* + \frac{\eta}{2p}\|\mathbf{g}_t - \nabla f(\mathbf{x}_t)\|^2$. Combining eq. (15) and eq. (16), we have

$$\mathbb{E}\left[\Phi_{t+1}\right] = \mathbb{E}\left[f(\mathbf{x}_{t+1}) + \frac{\eta}{2p}\|\mathbf{g}_{t+1} - \nabla f(\mathbf{x}_{t+1})\|^2\right]$$

$$\le \mathbb{E}\left[f(\mathbf{x}_t) - \frac{\eta}{2}\|\nabla f(\mathbf{x}_t)\|^2 - \frac{\eta}{2}\|\mathbf{g}_t - \nabla f(\mathbf{x}_t)\|^2 - \left(\frac{1}{2\eta} - \frac{l}{2}\right)\|\mathbf{x}_{t+1} - \mathbf{x}_t\|^2\right]$$

$$+ \frac{\eta}{2p}\mathbb{E}\left[p\epsilon^2 + (1 - p)\|\mathbf{g}_t - \nabla f(\mathbf{x}_t)\|^2 + (1 - p)\frac{l^2 \epsilon^{2/3}}{\sigma^{2/3}}\|\mathbf{x}_{t+1} - \mathbf{x}_t\|^2\right] \qquad (17)$$

$$\le \mathbb{E}\left[\Phi_t\right] - \frac{\eta}{2}\|\nabla f(\mathbf{x}_t)\|^2 - \underbrace{\left(\frac{1}{2\eta} - \frac{l}{2} - \frac{\eta(1-p)}{p} \cdot \left(\frac{l^2 \epsilon^{2/3}}{\sigma^{2/3}}\right)\right)}_{B}\|\mathbf{x}_{t+1} - \mathbf{x}_t\|^2 + \frac{\eta \epsilon^2}{2}.$$

Since we have chosen $\eta = \frac{1}{2l}$ and $p = \epsilon^{2/3}\sigma^{-2/3}$, it holds that

$$B \ge \frac{l}{2}\left(1 - \frac{\epsilon^{2/3}}{p\sigma^{2/3}}\right) \ge 0.$$

Thus we have:

$$\frac{1}{T}\sum_{i=0}^{T-1} \mathbb{E}\left[\|\nabla f(\mathbf{x}_i)\|^2\right] \le \frac{2}{\eta T}\mathbb{E}\left[\Phi_0 - \Phi_T\right] + \frac{\epsilon^2}{2}$$

$$\le \frac{2}{\eta T}\mathbb{E}\left[f(\mathbf{x}_0) - f(\mathbf{x}_T) + \frac{\eta}{p}\|\mathbf{g}_0 - \nabla f(\mathbf{x}_0)\|^2\right] + \frac{\epsilon^2}{2}$$

$$\le \epsilon^2,$$

where the last inequality is by setting

$$T = \left\lceil 8l\Delta\epsilon^{-2} + 4\sigma^{2/3}\epsilon^{-4/3}\right\rceil.$$

In the following, we bound the total queries of quantum stochastic gradient oracles. The expectation oracles to construct $\hat{\mathbf{g}}_t$ is

$$b_0 = \tilde{\mathcal{O}}\left(\sqrt{d}\sigma\hat{\sigma}_{1,t}^{-1}\right) = \tilde{\mathcal{O}}\left(\sigma\sqrt{d}\epsilon^{-1}\right),$$

and the expectation queries to construct $\Delta \mathbf{g}_t$ is

$$b_1 = \tilde{\mathcal{O}}\left(\sqrt{d}l\|\mathbf{x}_t - \mathbf{x}_{t-1}\|\hat{\sigma}_{2,t}^{-1}\right) = \tilde{\mathcal{O}}\left(\sqrt{d}\sigma^{1/3}\epsilon^{-1/3}\right).$$

Thus, the total stochastic quantum gradient oracles for finding the $\epsilon$-stationary point of $f(\cdot)$ can be bounded by

$$T(b_0 p + (1-p)b_1) = \tilde{\mathcal{O}}\left(\sqrt{d}(l\Delta\sigma^{1/3}\epsilon^{-7/3} + \sigma\epsilon^{-5/3})\right).$$

$\square$

*Remark* G.2. QGM+ (Algorithm 3) improves the quantum stochastic gradient oracle of Q-SPIDER ([48, Algorithm 7]) by a factor of $\epsilon^{-1/6}$.

