# OpenReview forum: "Quantum Algorithms for Non-smooth Non-convex Optimization"
_NeurIPS.cc/2024/Conference — NeurIPS 2024 poster_

### Official Review · Reviewer_RNco · 2024-07-01

**Soundness:** 3
**Presentation:** 3
**Contribution:** 3
**Rating:** 6
**Confidence:** 1

**Summary:**

This paper considers using quantum methods for stochastic optimization, using zeroth order queries. It looks like the main idea is that using quantum methods, one can summarize over finite difference calculations quickly and efficiently, to arrive at approximate subgradients efficiently; this would usually be very inefficient for classical methods. Overall, they are able to show speedup, from $O(\epsilon^{-4})$ to $O(\epsilon^{-3})$.

**Strengths:**

The problem is well contained and the premise is believable.

The classical optimization bits looks reasonable, and the results make sense.

I skimmed through the appendix, and the classical optimization parts are reasonable.

**Weaknesses:**

Section 3 is a bit hard to follow. The specific speedup offered by the quantum method is not entirely clear, though it is likely coming from Theorem B1. Perhaps a deeper discussion of this, and why this quantum speedup exists (e.g. is it a consequence of Deusch Josza? Can you provide a more complete argument for where the speedup appears? )

**Questions:**

(minor) Why do you say that finding a Clarke subdifferential is harder than finding a smooth differential? Generally speaking the complexities are comparable.

**Limitations:**

no societal limitations

---

> ### Author Rebuttal · Authors · 2024-08-06
>
> We thank the reviewer for the positive and helpful comments.
>
> **To Weakness 1 (the presentation of Section 3)**:
>
> Thank you for this question. Hopefully, the following explanation will give you a better picture of the structure of Section 3.
>
> The main goal in Section 3 is to construct unbiased quantum estimators $\hat{{\bf g}}$ and ${\Delta}{\bf g}$, which will be used in Algorithms 1 and 2. We present the query complexities on ${\bf U}\_F$ in constructing these oracles in Section 3.2, Theorem 3.4.
>
> The implementation of such (mini-batch) unbiased quantum estimators $\hat{{\bf g}}$ and ${\Delta}{\bf g}$ requires the access to quantum stochastic oracles ${\bf O}\_{{\bf g}\_\delta}$ or ${\bf O}\_{\Delta{\bf g}\_\delta}$, which are not given.
> Instead, we only have access to the quantum stochastic function value oracle $\mathbf{U}\_F$. We show the procedure for constructing ${\bf O}\_{{\bf g}\_\delta}$ and  ${\bf O}\_{\Delta{\bf g}\_\delta}$ by ${\bf U}\_F$ and a quantum sampling oracle $\mathbf{O}\_{\xi, \mathbf{w}}$ in Lemma 3.2 and Corollary 3.3 in Section 3.1, respectively. Then, in Section 3.3, we explicitly show how this quantum sampling oracle $\mathbf{O}\_{\xi, \mathbf{w}}$ can be constructed from scratch.
>
>  A diagram showing the logic flow presented above is as follows:
> \begin{align}
> \underbrace{\hbox{Section 3.3}}\_{\text{Construct ${\bf O}\_{\xi,{\bf w}}$}} \longrightarrow \underbrace{\hbox{Lemma 3.2 and Corollary 3.3}}\_{\text{Obtain ${\bf O}\_{{\bf g}\_{\delta}}$ and ${\bf O}\_{\Delta{\bf g}\_{\delta}}$ from ${\bf O}\_{\xi,{\bf w}}$ and ${\bf U}\_F$}} \longrightarrow \underbrace{\hbox{Theorem 3.4}}\_{\text{Construct $\hat{{\bf g}}$ and $\Delta{{\bf g}}$ from ${\bf O}\_{{\bf g}\_{\delta}}$ and ${\bf O}\_{\Delta{\bf g}\_{\delta}}$}}
> \end{align}
>
>
> **To Weakness 2 (discussion on the quantum speedup)**:
>
> The quantum mean estimator (Theorem B.1) is one of the main ingredients providing the speedup which leads to better estimators of $\nabla f\_{\delta}({\bf x})$ and $\nabla f\_{\delta}({\bf y})-f\_{\delta}({\bf x})$ as presented in Theorem 3.4 (also see the discussion in Remark 3.5).
> However, achieving the speedup shown in Theorem 3.4 requires additional considerations, including implementing a sampling oracle for the desired distribution and constructing a stochastic gradient oracle from it and the function value oracle. A high-level picture of those steps is detailed above in our answer to Weakness 1.
>
> Incorporating such better estimators in our quantum algorithm allows us to achieve better query complexities over the classical ones.
> More concretely, one needs to carefully select the variance level $\hat{\sigma}\_{1,t}^2$ in Algorithm 1 and $\hat{\sigma}\_{1,t}^2, \hat{\sigma}\_{2,t}^2$ in Algorithm 2 when utilizing the Theorem 3.4.
>
> As for the speedup on the quantum mean estimator [13, 42] over classical estimators, it can be viewed as a consequence of combinations of quantum Fourier transform [A] and the quantum amplitude amplification algorithm [B].
>
> We will involve more discussion on this in the revision.
>
> [A]. Shor, P. W. “Algorithms for quantum computation: Discrete logarithms and factoring”. In: Proceedings of the 35th Annual Symposium on Foundations of Computer Science, 1995.
>
> [B]. G. Brassard, P. Høyer, M. Mosca, and A. Tapp. “Quantum Amplitude Amplification and Estimation”. In: Contemporary Mathematics 305, 2002.
>
> **To Quetsion 1**:
> The complexities of finding Clarke differential and smooth differential are not comparable due to the following reasons:
>
> 1. We do not assume $f(\cdot)$ is differential. Since $f(\cdot)$ it in general non-convex and non-smooth, thus the differential of it may be intractable.
>
> 2. There are no finite time algorithm that can find an $\epsilon$-stationary point such that $\|\|\partial f({\bf x})\|\|\leq \epsilon$ in non-convex and non-smooth setting, where $\partial f(\cdot)$ denotes the Clarke differential of $f(\cdot)$. Please refer to [Theorem 5, 51].
> On the other hand finding an $\epsilon$-stationary point of a smooth differential in non-convex smooth optimization can be done in polynomial time [18, 31].

---

### Official Review · Reviewer_mFHz · 2024-07-11

**Soundness:** 4
**Presentation:** 3
**Contribution:** 3
**Rating:** 6
**Confidence:** 4

**Summary:**

This paper investigates quantum algorithms for finding the $(\delta,\epsilon)$-Goldstein stationary point of a potentially nonconvex and nonsmooth objective function $f$. Utilizing quantum variance reduction techniques as outlined in [42], the authors have developed a zeroth-order quantum estimator for the gradient of the smoothed surrogate of $f$. The stationary point of this smoothed surrogate is also the Goldstein stationary point of $f$ when using an appropriate smoothing parameter $\delta$. Leveraging this zeroth-order quantum estimator, the authors propose two algorithms, QGFM and QGFM+, to find the Goldstein stationary point, achieving a quantum speedup on the order of $\epsilon^{-2/3}$. Additionally, the QGFM+ framework adjusts the variance level during each variance reduction step, providing further acceleration to the Q-SPIDER algorithm described in [42] for smooth nonconvex optimization.

**Strengths:**

This paper initiates the study of quantum algorithms for finding Goldstein stationary points, a significant problem in continuous optimization. Additionally, the authors present an explicit construction of the quantum sampling oracle using the quantum zeroth-order oracle, including a detailed discussion on the number of qubits required.

**Weaknesses:**

Despite the detailed implementation and calculations, the overall technical approach remains relatively straightforward. The zeroth-order quantum estimator combines the classical stochastic gradient estimator for the smoothed surrogate with the quantum variance reduction algorithm in [42]. The quantum algorithms for finding the Goldstein stationary point are obtained by replacing the classical estimators with quantum estimators. Moreover, the narrative is somewhat incomplete due to the absence of lower bound results.

**Questions:**

Is it possible to improve the $\delta$ dependence using quantum algorithms?

Minor issues:

1. Consistency of big-O notation. For example, $O$ is used in line 139 and $\mathcal{O}$ in line 183. Similarly, there are consistency issues with the quantum oracle notation, where $\mathcal{O}$ is used in line 168 and $\mathbf{O}$ in line 184.

2. Typo on the RHS of the inequality in line 125.

3. The use of dashes '-' is a bit odd. For example, the dashes in line 139, line 210, and line 251 can be removed.

4. The name initials in the citation format are not precise. For example, in entry [1], it should be 'G. Arfken' instead of 'G Arfken'.

5. Line 310: "adjust" -> "adjusts". Line 311: "fixed" -> "fixes".

---

> ### Author Rebuttal · Authors · 2024-08-06
>
> We thank the reviewer for the positive and helpful comments.
>
> **To Weakness 1 (discussion on the technical novelty):**
>
> We appreciate the feedback. We highlight our technical novelty on the construction of zeroth-order estimator and the design of quantum algorithms as follow:
>
> * In term of the zeroth-order quantum estimator, utilizing the existing quantum mean value estimation procedure requires having a suitable quantum sampling oracle that returns a superposition over the desired distribution.
> We fixed the following gaps between this paper and the one of [42]:
>    1. [42] requires the assumption of direct accesses of quantum stochastic gradient oracle.
> But this is not applicable in our setting because instead, we are given only access to quantum stochastic function value oracle ${\bf U}\_F$.
> We overcome this by providing the efficient procedure for constructing our wanted oracle from ${\bf U}\_F$ and sampling oracle $\bf{O}\_{\xi, {\bf w}}$, which is summarized in Lemma 3.2.
>
>    2. Furthermore, [42] does not provide how to construct the quantum sampling oracle, which leaves a gap between the quantum algorithm and its detailed implementation on the quantum circuits. We fill this gap by giving an explicit and efficient construction on ${\bf O}\_{\xi,{\bf w}}$ for desired distribution in Section 3.3.
>
> * In term of the quantum algorithms for finding the Goldstein stationary point, our new proposed quantum algorithms are quite different from the classical ones for finding the Goldstein stationary point and the quantum methods for non-convex optimization.
>
>    1.  The most related classical algorithm is GFM+[7], where they adopted SPIDER algorithm, while we adopt the PAGE-framework, which makes the algorithm single-loop. To the best of our knowledge, such framework has not been investigated in finding the Goldstein stationary point even for classical optimization.
>     2. Our algorithm framework is also different from existing quantum algorithms for non-convex optimization, where they fixed the desired variance level when using quantum mean estimator, while we adaptively set the variance level and make $\hat{\sigma}_{2,t}\propto \|\|{\bf x}_t-{\bf x}\_{t-1}\|\|$, such strategy allows us not only to show the quantum advantage for finding the Goldstein stationary point in non-convex non-smooth optimization, but also to provide a better query complexity over the state-of-the-art non-convex smooth quantum optimization methods [42].
>
> **To Weakness 2 (Discussion on the Lower Bound)**:
>
> We agree that including a result of quantum lower bound will make the investigation on this problem more well-rounded. But we don't think the lack of a lower bound would diminish our contribution for the following reasons:
>
> 1. The main purpose of this paper is to show the quantum advantage on non-convex non-smooth optimization.
> We have provided an upper bound of $\tilde{\mathcal{O}}(d^{3/2}\delta^{-1}\epsilon^{-7/3})$ for finding the $(\delta,\epsilon)$-Goldstein stationary point,
> which cannot be achieved by ANY of the classical methods due to the classical lower bound $\delta^{-1}\epsilon^{-3}$ established by [14, 30].
>
> 2. Even for non-convex smooth function, the quantum lower bound has not been fully investigated [42, 49].
> There are only negative results that show no quantum speed-up for non-convex smooth optimization when the dimension $d$ is large [49].
> On the other hand, the construction of classical lower bound for finding Goldstein stationary point is based on the lower bound of finding stationary point of non-convex smooth function [14, 30].
> Hence, we think the quantum lower bound can be regarded as an important open problem for both non-convex non-smooth optimization and non-convex smooth optimization.
>
> **To Question 1**:
>
> Existing zeroth-order methods for finding the $(\delta,\epsilon)$-Goldstein stationary point of a non-convex non-smooth objective are to find the $\epsilon$-stationary point [7, 32, 33] or $(\delta,\epsilon)$-stationary point [29] of its smoothed surrogate $f_{\delta}({\bf x})$.
> However, there is a gap between the function value of $f(\cdot)$ and $f_{\delta}(\cdot)$, which can be only bounded by $|f(\cdot)-f_{\delta}(\cdot)|\leq \delta L$. This yields a factor of $\delta^{-1}$ in iteration complexities, which is irrelevant to using classical or quantum oracles.
>
> Therefore, we think the improvement on $\delta$ dependency cannot be achieved by existing algorithm framework and leave it as an important future work.
>
> **To Minor Issues 1-5**:
> Thanks for pointing out these. We will make the consistency of the big-O notation in the revision and fix the typos.

---

> > ### Comment · Reviewer_mFHz · 2024-08-11
> >
> > Thank you to the authors for their detailed response. I remain my rating

---

### Official Review · Reviewer_hDMr · 2024-07-12

**Soundness:** 3
**Presentation:** 3
**Contribution:** 3
**Rating:** 7
**Confidence:** 3

**Summary:**

This paper studies quantum algorithm for non-smooth non-convex stochastic optimization with zeroth-order oracle. It introduces an effective quantum estimator that reduces the variance compared to classical zeroth-order estimators. Upon substituting this estimator into known zeroth-order non-smooth optimizers, namely GFM and GDM+, the resulting quantum optimizer achieves improved rate $\tilde O(d^{3/2}\delta^{-1}\epsilon^{-3})$ and $\tilde O(d^{3/2}\delta^{-1}\epsilon^{-7/3})$ respectively for finding a $(\delta,\epsilon)$-Goldstein stationary point. Notably, quantum speedup improves upon the classical lower bound $\delta^{-1}\epsilon^{-3}$ by a factor of $\epsilon^{2/3}$. Moreover, a modified algorithm achieves $O(\sqrt{d}\epsilon^{-7/3})$ for smooth optimization, improving upon the best known rate.

**Strengths:**

This paper proposes a new zeroth-order quantum estimator. This leads to new quantum algorithms that solves zeroth-order non-smooth non-convex optimization problem, which is not well studied in the literature. Moreover, the proposed algorithms show quantum speedup compared to their classical (non-quantum) counterparts. Notably, it improves over the classical lower bound of $\Omega(\delta^{-1}\epsilon^{-3})$ by a factor of $\epsilon^{2/3}$. Overall, these results represent a significant contribution to the understanding of optimization with quantum oracles. Given my expertise lies primarily in optimization and not in quantum computation, I am only able to assess the optimization-related aspects of this work.

**Weaknesses:**

Although the dependence on $\delta,\epsilon$ is improved, the dimension dependence is suboptimal. In particular, since GFM and GFM+ are known to have suboptimal dimension dependence $d^{3/2}$, so do QGFM and QGFM+. On the other hand, as observed by Kornowsky and Shamir [1], optimizing the random smoothing $f_\delta$ with a non-smooth optimizer, such as online-to-non-convex (o2nc) [2], eliminates this $\sqrt{d}$ factor and achieves $O(d)$ in dimension. Hence, my intuition suggests that upon substituting the quantum estimator into o2nc and following a similar approach to Kornowsky and Shamir, the authors might be able to recover $O(d)$ (or even better) dimension dependence.

[1] Kornowski, G. and Shamir, O., “An Algorithm with Optimal Dimension-Dependence for Zero-Order Nonsmooth Nonconvex Stochastic Optimization”, 2023. doi:10.48550/arXiv.2307.04504.

[2] Cutkosky, A., Mehta, H., and Orabona, F., “Optimal Stochastic Non-smooth Non-convex Optimization through Online-to-Non-convex Conversion”, 2023. doi:10.48550/arXiv.2302.03775.

**Questions:**

- As someone unfamiliar with quantum computation, I have a general question: Is the proposed quantum oracle practically feasible to implement, or is it purely theoretical?

- line 87: does state $|i\rangle$ denote the $i$-th orthonormal basis of $\mathcal{H}^m$?

- line 100: what does it mean by $|\mathbf{x}\rangle |q\rangle$? Is it a shorthand for tensor product?

- Thm 3.4 part 1: should it be $Var(\hat g) \le \hat\sigma_1^2$ instead of $\hat \sigma_1$? part 2: number of queries should be $\frac{d^{3/2}L\\|y-x\\|}{\delta \hat\sigma_2}$ (i.e., currently it's missing $1/\delta$)?  Since this theorem is the main result of the quantum oracle, I encourage the authors to carefully check its correctness.

  also in the proof (line 471): $\sigma_1^2$ => $\hat\sigma_1^2$?

Minor comments:

- line 98: $C_{f(x)} = f(x)$ => $C_f(x) = f(x)$?
- Proposition 2.1: the properties of smooth surrogate $f_\delta$ are known in [1] and [2], and Lin et. al. and Chen et. al. are restating these results in their papers. Hence, these should be more appropriate references.

[1] Yousefian, F., Nedić, A., and Shanbhag, U. V., “On Stochastic Gradient and Subgradient Methods with Adaptive Steplength Sequences”, 2011. doi:10.48550/arXiv.1105.4549.

[2] Duchi, J. C., Bartlett, P. L., and Wainwright, M. J., “Randomized Smoothing for Stochastic Optimization”, 2011. doi:10.48550/arXiv.1103.4296.

---

> ### Author Rebuttal · Authors · 2024-08-06
>
> We thank the reviewer for the positive and helpful comments.
>
> **To Weakness 1**:
> We think such sub-optimal dependency on $d$ is reasonable due to the following reasons:
>
> 1. The sub-optimality on dimension $d$ is a common trade-off in quantum optimization. [49] proved that there are no quantum speedups for finding stationary points of a non-convex smooth object function when the dimension $d$ is large. [42] showed that it requires $\mathcal{O}(\sqrt{d}\epsilon^{-2.5})$ queries of quantum stochastic first-order oracle to find the $\epsilon$-stationary point of a non-convex smooth objective, while the classical optimal methods [18, 31] require $\mathcal{O}(\epsilon^{-3})$ queries. This paper considers a more difficult function class which is non-convex and non-smooth.
>
> 2. The algorithm frameworks pointed out by the reviewer in Kornowski and Sharmir [30], Cutkosky et al. [14] construct the gradient estimator by $2$-queries of stochastic function value oracle or one query of stochastic gradient oracle, which means we cannot apply quantum mean estimator to improve such estimator.
> Hence, whether using their framework can recover the $\mathcal{O}(d)$ dependency while still improve the dependency on $\epsilon$ is not clear.
>
> **To Question 1**:
> We answer this question from the following two aspects:
>
> 1. The quantum oracles can be viewed as instances of quantum circuits.
>     Since the current hardware implementation of quantum computers is still in a relatively early stage,
>     the practical feasibility of implementing these quantum circuits depends on factors such as the size of the circuits and the types of gates used.
>
> 2. The gates used in constructing our quantum stochastic gradient oracle, aside from the provided black-box quantum function value oracle, are primarily simple single-qubit and two-qubit gates, such as Hadamard gate, controlled NOT gate, and controlled rotation gate, which are known to be efficiently implementable [A].
> Therefore, if the size of our problem is not large and the black-box quantum function value oracle is practically feasible, then our algorithm can be practically feasible.
>
> [A]. Groenland, Koen, et al. "Signal processing techniques for efficient compilation of controlled rotations in trapped ions." New Journal of Physics, 2020.
>
> **To Question 2 and 3**:
> Yes.  Your understanding is correct and we will make them clear in the revision.
>
> **To Question 4**:
> Thanks for pointing out this.
> For part 1, it should be $\mathbb{E}\left[\||\hat{{\bf g}}-\nabla f\_{\delta}({\bf x})\||^2\right]\leq \hat{\sigma}\_1^2$, which is a typo.
> For part 2, it should be $d^{3/2}L\|\|{\bf y}-{\bf x}\|\|\hat{\sigma}_{2}^{-1}\delta^{-1}$, where we forget to present the dependency on $\delta$.
>
> They do not impact the results of Theorem 4.1 and Theorem 4.3.
> Part 2 of Theorem 3.4 only impact the size of $b_1$ in Theorem 4.3. We determine the size of $b_1$ in line 507 of Appendix D by setting
> $$
>     \hat{\sigma}\_{2,t}^2 = \epsilon^{-2/3}\|\|{\bf x}\_{t+1}-{\bf x}\_t\|\|^2 L^{4/3}d\delta^{-1}
> $$
> according to line 494, then we have
> \begin{align}
>     b\_1 = \tilde{\mathcal{O}}\left(d^{3/2}L\|\|{\bf x}\_{t+1}-{\bf x}\_t\|\|\hat{\sigma}\_{2,t}^{-1}\delta^{-1}\right)
>      = \tilde{\mathcal{O}}\left(d^{3/2}L\|\|{\bf x}\_{t+1}-{\bf x}\_t\|\| (\epsilon^{-1/3}\|\|{\bf x}\_{t+1}-{\bf x}\_t\|\|^{-1}L^{-2/3}d^{-1/2}\delta)\delta^{-1}\right)
>      = \tilde{\mathcal{O}}\left(dL^{1/3}\epsilon^{-1/3}\right),
> \end{align}
> which means $b_1$ remains the same and it is independent of $\delta$.
>
> We will fix them in the revision.
>
> **To Question 5 and 6**:
> Thanks for pointing out these, we will modify them in the revision.

---

> > ### Comment · Reviewer_hDMr · 2024-08-09
> >
> > Thank the authors for their detailed response. Regarding weakness, thanks for providing the additional background. Now I understand improving dimension dependence is a non-trivial challenge. Given this, I adjusted my score accordingly.
> >
> > As a quick follow-up question, the authors mentioned that quantum estimators can only estimate one-point stochastic function value $f(x,z)$, but not 2-point function values nor stochastic gradients. Is this limitation proven impossible in the quantum computing literature, or is it still an open problem?

---

> > > ### Author Response · Authors · 2024-08-09
> > >
> > > Thanks for your positive comments and raising your score.
> > > We present the answer for your follow-up question as follows:
> > >
> > > We do not mean that ‘’quantum estimators can only estimate one-point stochastic function value $f(x,z)$, but not 2-point function values nor stochastic gradients''.
> > > In fact, we can use quantum oracles to construct gradient estimators by $2$-queries of stochastic function value oracle (see our result in Theorem 3.2) or a query of stochastic gradient oracle [42].
> > >
> > > However, the speed-up by the quantum mean estimators requires to use mini-batch queries of stochastic function value oracles or mini-batch queries of stochastic gradient oracles to reduce oracle calls with a given variance level (see our Theorem 3.4 and Remark 3.5).
> > > Hence, whether one can construct the stochastic gradient estimators by $2$-queries (or constant queries) of stochastic function value oracle or a query of stochastic gradient oracle (as in the algorithms of [30, 41])  while still maintain the quantum speed-up remains an open problem.
> > >
> > > We hope this answers your following-up question and are happy to address any further concern.

---

> > > > ### Comment · Reviewer_hDMr · 2024-08-10
> > > >
> > > > Thank you to the authors for their thorough follow-up response. It has addressed all of my questions, and I have no further questions.

---

### Official Review · Reviewer_KzCk · 2024-07-13

**Soundness:** 3
**Presentation:** 2
**Contribution:** 3
**Rating:** 6
**Confidence:** 4

**Summary:**

This paper introduces new quantum algorithms for non-smooth non-convex optimization problems. The authors propose a quantum gradient estimator for smoothed objectives and develop the Quantum Gradient-Free Method (QGFM) and its enhanced version, QGFM+, which achieve better query complexities than their classical counterparts. These complexities demonstrate a marked quantum speedup over classical counterparts, indicating the potential of quantum computing in optimizing complex functions more efficiently. The paper also discusses the construction of quantum oracles and the application of variance reduction techniques, paving the way for future research in quantum optimization.

**Strengths:**

- The paper proposed new zeroth order quantum optimization algorithms achieving better computational complexities compared to classical methods for non-smooth and non-convex optimization.
- Technically, they construct efficient quantum gradient estimators and quantum superpositions over required distributions as a key subroutine.
- They also proposed a quantum algorithm for non-convex smooth problems with an adaptive variance level, accelerating prior quantum algorithms to get more speedups.

**Weaknesses:**

- The assumptions of having a quantum stochastic function value oracle may be strong. Could the authors explain more about why it is reasonable and important to have such a function oracle?
- The technical core for quantum speedups seems to be the quantum mean value estimation procedure, which is already used in many other optimization problems and scenarios. Could the authors explain more about the technical novelty of their work?

**Questions:**

Besides the questions raised in the weakness part, I have some minor issues with the submission as follows:

- In line 89, the definition of the tensor product may be a little confusing.
- In the explicit construction of quantum sampling oracles, it seems that the time complexity of the quantum algorithm may be much larger than the query complexity, due to the sampling on the unit sphere. However, for such optimization algorithms, time complexity may be more crucial in real-world applications. Could the authors state the actual time complexity of their algorithm in terms of gate counts?

**Limitations:**

The quantum complexity lower bound on this problem is not proved in this paper, which is mentioned in the conclusion part.  Also, as noted in remark 3.7, implementing quantum sample oracle may require the uses of QRAM, which is currently limited by the physical realizations.
This is a theoretical work, so there is no potential negative societal impact of their work.

---

> ### Author Rebuttal · Authors · 2024-08-06
>
> We thank the reviewer for the positive and helpful comments.
>
>  **To Weakness 1**:
> We think the assumption of having a quantum stochastic function value oracle is reasonable and not strong due to the following reasons:
>
> 1. It is very common to assume having a classical stochastic function value oracle, when the objective is in the form of expectation [7, 30, 32, 33].
>
> 2. One can efficiently construct such quantum stochastic function value oracle with the same asymptotic computational complexity by replacing the gates in the classical circuit with reversible gates [38].
>
> Such quantum stochastic function value oracle is important to further design quantum algorithms.
>
> **To Weakness 2**:
> Utilizing the existing quantum mean value estimation procedure requires having a suitable quantum sampling oracle that returns a superposition over the desired distribution.
> The closest work to ours is [42], which utilizes quantum mean estimation in non-convex optimization. We state below the technical novelty in our paper:
>
> 1.  [42] requires the assumption of direct accesses of quantum stochastic gradient oracle.
> But this is not applicable in our setting because instead, we are given only access to quantum stochastic function value oracle ${\bf U}_F$.
> We overcome this by providing the efficient procedure for constructing our wanted oracle from ${\bf U}_F$ and sampling oracle
> ${\bf O}\_{\xi, {\bf w}}$, which is summarized in Lemma 3.2.
>
> 2. Furthermore, [42] does not provide how to construct the quantum sampling oracle, which leaves a gap between the quantum algorithm and its detailed implementation on the quantum circuits.
> We fill this gap by giving an explicit and efficient construction on ${\bf O}\_{\xi,{\bf w}}$ for desired distribution in Section 3.3.
>
> In conclusion, our technical novelty on quantum part is to provide an explicit and efficient construction of quantum $\delta$-estimated stochastic gradient oracle (Definition 3.3) by using quantum stochastic function value oracle.
>
> **To Question 1**:
> We give a more specific description: Given $|{\bf x}\rangle \in \mathcal{H}^m$ and $|{\bf y}\rangle \in \mathcal{H}^n$, we denote their tensor product by $\ket{\bf x} \otimes  \ket{\bf y}\triangleq (x_1y_1, x_1y_2 \cdots, x_1y_n, x_2y_1, \cdots, x_m y_n)^\top \in \mathcal{H}^{m\times n}$.
>
> **To Question 2**:
> We want to first respectfully point out that it is inappropriate to compare computational complexity and query complexity.
> Computational complexity considers the number of basic operations or gates used, whereas query complexity measures the number of calls to a particular process in a black-box manner.
> In generally, this process can cause significant costs (including a large number of basic operations) or may not be efficiently computed locally.
>
> This paper focus on the query complexities, which follows the previous work in [4, 5, 42]. Unlike [42] which does not consider the implementation of the sampling oracles, our Section 3.3 enables us to give the gate counts for constructing one quantum $\delta$-estimated stochastic gradient oracle.
>
> Specifically, the steps in Lemma 3.2 needs $1$ access to sampling oracle, $1$ addition operation, $2$ subtraction operations, and $1$ multiplication operation. Hence, the gate complexity stems from the two parts below:
>
> 1. The gate complexity for implementing the sampling oracle is $\mathcal{O}(\lceil\log N\rceil + \log (1/\epsilon_0) d)$ where $N$ is the number of the stochastic components of $\xi$, and $\epsilon_0$ being the incurred precision error. More precisely, this construction utilizes $\lceil\log N\rceil + \log (1/\epsilon_0) d$ Hadamard gates and $d-1$ circuits of calculating $\sin$ and $\cos$ on a single qubit.
>
> 2. There are various existing methods to implement quantum arithmetic operations, e.g. [41]. Using methods from [41] gives implementations of addition, subtraction and multiplication with gate complexity $\mathcal{O}(C^2)$, $\mathcal{O}(C^2)$, and $\mathcal{O}(C^3)$, respectively, where $C$ is the number of qubits that represent the numbers being manipulated in our algorithm and is usually independent of $d$, $\epsilon$, and $\delta$.
>
> Hence, the asymptotic total gate complexity for constructing one quantum estimated gradient oracle is $\mathcal{O}(\lceil\log N\rceil+\log (1/\epsilon_0)  d + C^3)$.
>
> **To Limitation 1**:
> We agree that including a result of quantum lower bound will make the investigation on this problem more well-rounded. But we don't think the lack of a lower bound would diminish our contribution for the following reasons:
>
> 1. The main purpose of this paper is to show the quantum advantage on non-convex non-smooth optimization.
> We have provided an upper bound of $\tilde{\mathcal{O}}(d^{3/2}\delta^{-1}\epsilon^{-7/3})$ for finding the $(\delta,\epsilon)$-Goldstein stationary point,
> which cannot be achieved by ANY of the classical methods due to the classical lower bound $\delta^{-1}\epsilon^{-3}$ established by [14, 30].
>
> 2. Even for non-convex smooth function, the quantum lower bound has not been fully investigated [42, 49].
> There are only negative results that show no quantum speed-up for non-convex smooth optimization when the dimension $d$ is large [49].
> On the other hand, the construction of classical lower bound for finding Goldstein stationary point is based on the lower bound of finding stationary point of non-convex smooth function [14, 30].
> Hence, we think the quantum lower bound can be regarded as an important open problem for both the non-convex non-smooth optimization and the non-convex smooth optimization.
>
> **To Limitation 2**:
> Our algorithms **don't need** QRAM.
> Remark 3.7 serves as a discussion for general scenarios on the distribution of $\xi$.
> More concretely, a distribution that can be efficiently sampled classically guarantees an efficient construction of a quantum sample oracle over it; while for other cases, QRAM might be needed on a case-by-case basis.

---

> ### Comment · Reviewer_KzCk · 2024-08-10
>
> Thank the authors for the detailed response!
> Since the quantum upper bound provided in the paper is $\tilde{\mathcal{O}}(d^{3/2}\delta^{-1}\epsilon^{-7/3})$, while a possible classical upper bound is $\mathcal{O}(d\delta^{-1}\epsilon^{-3})$ the classical lower bound is $\delta^{-1}\epsilon^{-3}$, does it mean that quantum speedup can only occur when the requirement of the precision is relatively high compared with the dimension, given the dimension fixed? If so, regarding current physical realization constraints like noises in quantum devices, it seems that this result is more of a theoretical one with few potential applications.
> Another problem is the time/gate complexity part. Thank the authors for their effort to analyze and calculate the gate cost in great detail of constructing one quantum estimated gradient oracle. However, it seems that for one call of the gradient oracle, the cost is at least proportional to the dimension, which is relatively expensive for some problems and may be a major concern in solving them.
> Due to the above reasons, I'll keep my rating currently and keep in mind your detailed comments for future discussions.

---

> > ### Author Response · Authors · 2024-08-10
> >
> > Thanks for your response and the follow-up questions.  We present detailed answers as follows.
> >
> > > Since the quantum upper bound provided in the paper is $d^{3/2}\delta^{-1}\epsilon^{-7/3}$, while a possible classical upper bound is $d\delta^{-1}\delta^{-3}$ the classical lower bound is $\delta^{-1}\epsilon^{-3}$, does it mean that quantum speedup can only occur when the requirement of the precision is relatively high compared with the dimension, given the dimension fixed?
> >
> > Yes, the quantum speedup occurs when $d$ and $\epsilon$ satisfies that $d\leq \epsilon^{-4/3}$.
> > We want to point out it is very common to show the quantum speed-up under the case that the dimension is not too large in non-convex optimization.
> > For the smooth case, there is NO quantum speedup for finding $\epsilon$-stationary point with stochastic gradient inputs when the dimension is large such that $d\geq \epsilon^{-3}$ [Proposition D.12, A]. Besides, [41] presents a quantum upper bound of $\epsilon^{-5/2}\sqrt{d}$ for smooth non-convex optimization, where the quantum speedup only occurs when $d\leq \epsilon^{-1}$.
> > The function class considered in this paper can be non-smooth, which is more general than the previous works.
> > Thus we think the speed-up region $d\leq \epsilon^{-4/3}$ obtained in this paper is reasonable.
> >
> > > If so, regarding current physical realization constraints like noises in quantum devices, it seems that this result is more of a theoretical one with few potential applications.
> >
> > We would like to point out that the device's noise mentioned by you can be taken care of by quantum error correction code [B] such as surface code.
> > On the other hand, the problem's dimension $d$ is limited by the scale of existing quantum computers and thus cannot be very large.
> > Hence we think the quantum speedup region $d\leq \epsilon^{-4/3}$ is not that strict.
> >
> > Though the number of precise qubits is still limited by the current technique, the quantum community believes that getting more enough precise qubits is just a matter of time [C].
> >
> > > Another problem is the time/gate complexity part. Thank the authors for their effort to analyze and calculate the gate cost in great detail of constructing one quantum estimated gradient oracle. However, it seems that for one call of the gradient oracle, the cost is at least proportional to the dimension $d$, which is relatively expensive for some problems and may be a major concern in solving them. ''
> >
> > Thanks for acknowledging our effort!
> >
> > We think the $d$-dependency computation cost for one call of the quantum estimated gradient oracle is unavoidable and should not be regarded as a major concern.
> > Evaluating a function value at a $d$-dimension input will at least take $\mathcal{O}(d)$ time to even just read the input.
> > Hence, our construction of the quantum estimated gradient oracle doesn't introduce much more cost than necessary, and it is natural to focus on the query complexities on stochastic function value oracle, which usually is the dominant part in the computation.
> > For instance, even for a simple function with the form $f({\bf x}) = {\bf x}^{\top}{\bf A}({\bf x}){\bf x}$, where ${\bf A}({\bf x})\in R^{d\times d}$, its computation cost of a given $\bf x$ will be at least proportional to $d^2$, which is much larger than $\mathcal{O}(d)$ for constructing one quantum estimated gradient oracle.
> >
> > **References:**
> >
> > [A]. Chenyi Zhang, and Tongyang Li. Quantum Lower Bounds for Finding Stationary Points of Nonconvex Functions. ICML, 2023.
> >
> > [B]. Roffe J. Quantum error correction: an introductory guide. Contemporary Physics, 2019.
> >
> > [C] Gambetta, Jay M., Jerry M. Chow, and Matthias Steffen. Building logical qubits in a superconducting quantum computing system. npj quantum information, 2017.
> >
> > We hope our response can address your concern and are happy to answer any further questions.

---

> > > ### Comment · Reviewer_KzCk · 2024-08-11
> > >
> > > Thank you for the detailed response! It has addressed all of my questions, and I adjusted the score accordingly.

---

### Decision · Program_Chairs · 2024-09-25

**Decision:**

Accept (poster)

**Comment:**

This paper studied quantum algorithms for non-smooth non-convex optimization. Classical algorithms for such optimization problems have been relatively well-studied, but it is widely open in the setting of quantum computing and the paper initiated such studies.

The novelty and technical contribution are well-received by the reviews, and during the discussion period the authors addressed concerns raised from reviewers. The scores are unianimously positive and the decision is to accept this paper at NeurIPS 2024.

PS. For the final version, the authors should carefully incorporate all the points raised during rebuttal. In particular, the authors should take a close look at Review RNco, especially the complexity of subgradients of non-smooth function. It would be helpful to clarify that the objective of your paper is to explore algorithms with good worst-case bounds, but you could make a statement clarifying that for specific real-world problems, more efficient methods may (and often do) exist.